# Self-Evaluation Guided Beam Search for Reasoning

**Yuxi Xie**[1*]   **Kenji Kawaguchi**[1]   **Yiran Zhao**[1]   **James Xu Zhao**[1]
**Min-Yen Kan**[1†]   **Junxian He**[2†]   **Michael Qizhe Xie**[1†]

[1] National University of Singapore [2] The Hong Kong University of Science and Technology

## Abstract

Breaking down a problem into intermediate steps has demonstrated impressive performance in Large Language Model (LLM) reasoning. However, the growth of the reasoning chain introduces uncertainty and error accumulation, making it challenging to elicit accurate final results. To tackle this challenge of uncertainty in multi-step reasoning, we introduce a stepwise self-evaluation mechanism to guide and calibrate the reasoning process of LLMs. We propose a decoding algorithm integrating the self-evaluation guidance via stochastic beam search. The self-evaluation guidance serves as a better-calibrated automatic criterion, facilitating an efficient search in the reasoning space and resulting in superior prediction quality. Stochastic beam search balances exploitation and exploration of the search space with temperature-controlled randomness. Our approach surpasses the corresponding Codex-backboned baselines in few-shot accuracy by $6.34\%$, $9.56\%$, and $5.46\%$ on the GSM8K, AQuA, and StrategyQA benchmarks, respectively. Experiment results with Llama-2 on arithmetic reasoning demonstrate the efficiency of our method in outperforming the baseline methods with comparable computational budgets. Further analysis in multi-step reasoning finds our self-evaluation guidance pinpoints logic failures and leads to higher consistency and robustness. Our code is publicly available at https://guideddecoding.github.io/.

## 1 Introduction

 The remarkable empirical achievements of Large Language Models (LLMs) have recently ushered in a new era in machine reasoning through few-shot prompting techniques (Brown et al., 2020; Chowdhery et al., 2022; Touvron et al., 2023a; OpenAI, 2023). In particular, breaking down a problem into intermediate stages, or a reasoning chain, can significantly improve model performance on reasoning tasks (Cobbe et al., 2021). Various prompting approaches have been proposed to define these chains, such as *scratchpads* (Nye et al., 2021), *chain-of-thought* (CoT) (Wei et al., 2022b), *least-to-most* (Zhou et al., 2023), and *program-aided language models* (PAL) (Gao et al., 2023; Chen et al., 2022). However, as the complexity and length of reasoning chains increase with the difficulty of tasks, LLMs struggle with errors and imperfections that accumulate across multiple intermediate steps (Wu et al., 2016; Guo et al., 2018; Chen et al., 2022). Furthermore, the growing number of steps leads to an exponential growth in the search space for reasoning, making it exceedingly difficult to obtain accurate final outcomes.

Confronted with the challenges of uncertainty in multi-step chaining, several previous studies have worked on different aspects to alleviate the impact of reasoning errors. For instance, Wang et al. (2023) introduce *self-consistency* as a method to determine the final answer through majority voting using multiple sampled reasoning paths, while Li et al. (2022) investigate various prompts to diversify the sampling outcomes. Gao et al. (2023) and Chen et al. (2022) utilize Python programs to

---

[*]Correspondence to: Yuxi Xie (xieyuxi@u.nus.edu).
[†]Equal advising. Ordering is determined by dice rolling.

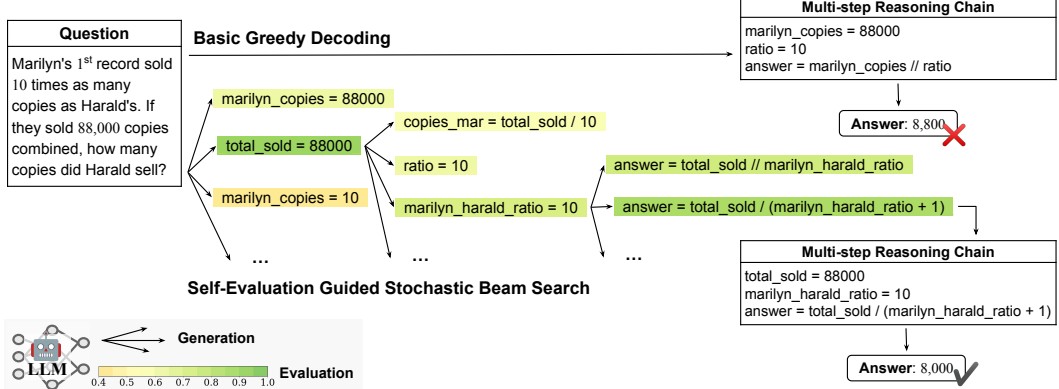

Figure 1: Self-Evaluation can calibrate the decoding direction in multi-step reasoning. We illustrate our method in the form of stepwise stochastic beam search with the beam size equal to 1. The scale of the self-evaluation score is visualized in the colormap. We adopt Program-Aided Language models (PAL) reasoning (Gao et al., 2023; Chen et al., 2022) for this math word problem.

achieve higher accuracy in mathematical computations. While these approaches have contributed to significant performance improvements in reasoning, the process of generating reasoning chains has been parameterized as a standard autoregressive process and intrinsically faces the challenge of sampling within an exponentially large search space.

Motivated by this challenge, we employ LLM *self-evaluation* (Kadavath et al., 2022) as a better-calibrated criterion to automatically guide the search in the reasoning space, drawing inspiration from prior works on utilizing LLMs for self-evaluation (Rae et al., 2021; Paul et al., 2023; Madaan et al., 2023; Shinn et al., 2023). We integrate the self-evaluation guidance for reasoning in a stepwise and generalizable manner. Specifically, we formulate the reasoning chain generation as a decoding process consisting of multiple intermediate steps. Unlike traditional text decoding where each step produces a single token, we consider each decoding step as a reasoning logic composed of a sequence of tokens. This framework enables us to employ beam search (Jurafsky and Martin, 2009; Graves, 2012) decoding tailored for intermediate steps and guide the beam searching process by controlling the error of each reasoning step to prevent potential error accumulation throughout the chaining. Figure 1 illustrates an example of decoding a chain of program-aided reasoning steps. Furthermore, we incorporate temperature-controlled randomness (Ackley et al., 1985; Kool et al., 2019; Meister et al., 2021) into the traditional (deterministic) beam search to balance the quality–diversity trade-off in searching for better reasoning chains. Our approach has resulted in respectable improvements across various arithmetic, symbolic, and commonsense reasoning tasks. For instance, by guiding the reasoning decoding process of the Codex model (Chen et al., 2021), we achieve accuracies of 85.5%, 64.2%, and 77.2% on the GSM8K, AQuA, and StrategyQA benchmarks, compared to the vanilla reasoning-enhanced Codex performance of 80.4%, 58.6%, and 73.2%, respectively. Our further analysis on Llama-2 (Touvron et al., 2023b) demonstrates the efficiency of our method in surpassing the self-consistency baseline under equivalent computational budgets.

## 2 Self-Evaluation Guided Stochastic Beam Search

Considering the input prompt and question $Q$ represented as $x$, we formulate the answer distribution $P(a \mid x)$ by decomposing it as a reasoning chain generation process $P(R \mid x)$ and an answer generation process $P(a \mid R, x)$:

$$P(a \mid x) = \mathbb{E}_{R \sim P(R|x)} P(a \mid R, x), \tag{1}$$

where $R$ is the intermediate reasoning chain variable that is typically a text sequence. $P(a \mid R, x) = \frac{\mathbb{1}_A(a)}{\max(|A|, 1)}$, where $A = \texttt{execute}(R)$ represents the set of predicted answer(s) interpreted from $R$, and $\mathbb{1}_A$ is the indicator function of the subset $A$. In practice, $|A| \geq 0$ can be 0 or larger than 1 when the reasoning $R$ returns no valid answer or produces more than one possible answers, respectively.

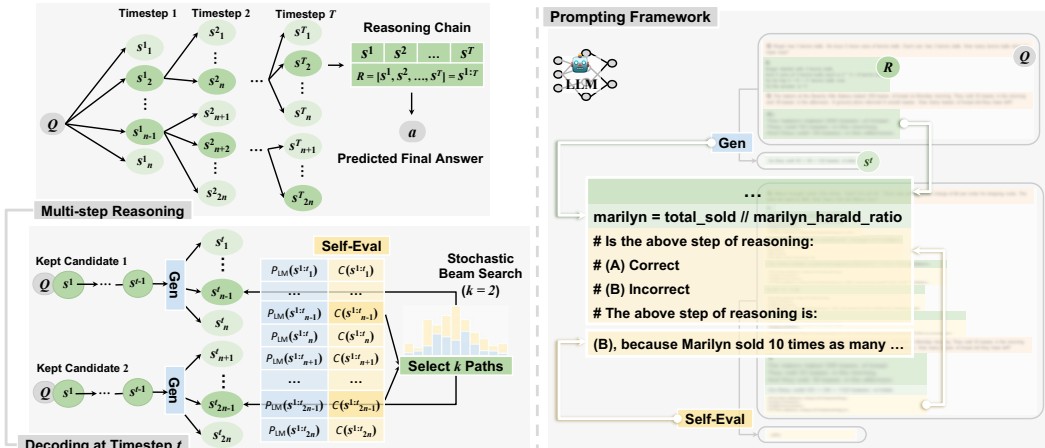

Figure 2: Our framework of self-evaluation guided stochastic beam search for multi-step reasoning. The schema of the decoding process is on the left, where we keep $k = 2$ candidates at each timestep, with the detailed illustration of timestep $t$ at the bottom. Here "Gen" and "Self-Eval" represent the generation and evaluation LLMs, respectively. The corresponding prompt formulations are provided on the right, where the questions $Q$, reasoning steps $R$, and evaluation scripts are highlighted in orange, green, and yellow, respectively. Steps in light green (*e.g.*, $s^t$) are for models to generate or evaluate at the current timestep. Specifically, we follow Kadavath et al. (2022) to prompt the LLM evaluation by answering the multiple-choice question, *i.e.*, the lines starting with #.

Prior research has modeled the reasoning chain generation $P(R \mid x)$ by prompting LLMs to explicitly elaborate on the required intermediate steps $R$. Through setting different prompting schemes, the reasoning process $P(R \mid x)$ can be modeled as chain-of-thought free-text reasoning (Kojima et al., 2022; Wei et al., 2022b), a two-stage question decomposition and answering pipeline (Zhou et al., 2023), or program-aided reasoning to generate a python program (Gao et al., 2023; Chen et al., 2022). While effective, previous work mostly uses a single sample of $R$ from the LLMs to approximate the expectation in Eq. 1 – the generated reasoning chain is often unreliable and causes incorrect answers. To mitigate this issue, Wang et al. (2023) conduct majority voting to approximate the expectation via sampling and aggregating multiple reasoning chains. Li et al. (2022) take a further step to diversify the sampling and calibrate $P(R \mid x)$ with a task-specific fine-tuned verifier. Another line of work focuses on improving $P(a \mid R, x)$ instead. For example, Gao et al. (2023) and Chen et al. (2022) employ Python programs for more accurate calculations in math word problems.

In this work, we focus on improving $P(R \mid x)$ to enhance the consistency of the sampled reasoning chains. To this end, we propose to explicitly break down the reasoning process into multiple steps, as shown in Figure 2, where each step yields a semantically integrated sequence of tokens, representing a single step within the overall reasoning chain. From this perspective, we can approach the task of enhancing $P(R \mid x)$ as a decoding problem over the reasoning chains. Considering the exponentially large search space and the potential unreliability of LLM-produced chains in reasoning, we propose a constrained stochastic beam search decoding approach to improve the reasoning step by step and obtain high-quality reasoning with a limited number of samples. We detail our approach next.

## 2.1 Multi-step Reasoning via Stochastic Beam Search

In multi-step reasoning, a reasoning chain of $T$ steps is sequentially generated through several timesteps as $R = [s^1, s^2, \cdots, s^T] = s^{1:T}$, where $s^t$ represents a sequence of tokens as the $t$-th step. Formally, the reasoning generation process $P(R \mid x)$ can be factorized in an autoregressive manner:

$$P(R = s^{1:T} \mid x) = \prod_t P(s^t \mid x, s^{1:t-1}), \tag{2}$$

which resembles the typical token-level autoregressive distribution of language models. Stepwise reasoning allows us to formulate the process as a step-by-step decoding problem, where we can utilize widely-used strategies such as beam search for the generation. Different from the typical text

decoding process where each step consists of a single token, here we view a sequence of reasoning tokens as a single step. One of the most severe issues in LLM-based reasoning is the potential unreliability and inaccuracy of each reasoning step generated by the model. Furthermore, errors from individual steps may accumulate throughout the reasoning chain, exacerbating the problem. To address the issue, we define a constraint function $\mathcal{C}(s^t, s^{1:t-1}) \in [0, 1]$ within each reasoning step[3] that outputs the LLM confidence in the correctness of the reasoning sequence $s^t$ based on the previous context $s^{1:t-1}$. Then, we present a constrained decoding approach that combines the language model probability and the correctness confidence as a new decoding objective function $\mathcal{E}(s^{1:T})$:

$$\mathcal{E}(s^{1:T}) = \prod_t P_{\mathrm{LM}_\mathcal{G}}^\lambda(s^t \mid x, s^{1:t-1})\mathcal{C}^{1-\lambda}(s^t), \tag{3}$$

where $P_{\mathrm{LM}_\mathcal{G}}$ is the language model distribution [4]. $\lambda \in [0, 1]$ is a weight hyperparameter to balance the LM score and the confidence score. We will detail the design of $\mathcal{C}(s^t)$ in Section 2.2. Eq 3 follows an autoregressive factorization form, and thus traditional token-level decoding methods such as beam search can be applied here on the chain level. As it is desirable to obtain high-quality reasoning chains with limited samples that are scored high by $\mathcal{E}(s^{1:T})$, it is natural to utilize greedy or beam search decoding to approximate the reasoning sequences that maximize $\mathcal{E}(s^{1:T})$.

Additionally, multiple diverse reasoning chains could be aggregated to further improve the final accuracy, as suggested by Eq 1 and empirically confirmed by self-consistency reasoning (Wang et al., 2023). To this end, we propose a variant of stochastic beam search (Kool et al., 2019; Meister et al., 2021) to strike a tradeoff between exploration and exploitation. Concretely, for beam size $k$, at each reasoning step we draw $n$ samples of $s^t$ following $P_{\mathrm{LM}_\mathcal{G}}(s^t \mid x, s^{1:t-1})$ for each beam, and we end up with $nk$ chain hypotheses of $s^{1:t}$ to form the candidate set $\mathcal{S}$, then we perform beam pruning through sampling – we sample $k$ reasoning beams without replacement, rather than finding the $\arg\max k$, following a distribution defined by the accumulated score:[5]

$$P_{beam}(s^{1:t}) \propto \exp(\mathcal{E}(s^{1:t})/\tau), \quad s^{1:t} \in \mathcal{S} \tag{4}$$

where the temperature $\tau$ is a hyperparameter to control the randomness in stochastic beam search; when $\tau \to 0$, stochastic beam search becomes the vanilla beam search algorithm. The reasoning beams $s^{1:t}$ can be sampled efficiently since $|\mathcal{S}| = nk$ is a finite set. To enable fine-grained control of sampling randomness in decoding, we also introduce a hyperparameter $\alpha \in [0, 1]$ so that $\tau$ can decay step by step as $\tau \to \alpha\tau$. By annealing $\tau$ with $\alpha$, we can mitigate the error accumulation due to aggregated randomness throughout chaining, as discussed in Section 3.4.

By incorporating controllable randomness, we not only achieve a more reliable single reasoning chain generation by setting randomness to be small, but also leverage multiple diverse reasoning chains with larger variance. Next, we introduce our constraint function $\mathcal{C}(s^t, s^{1:t-1})$ that utilizes a self-evaluation scheme to improve the consistency of each reasoning step.

## 2.2 Self-Evaluation as Correctness Control

Inspired by the recent success of self-evaluation (Kadavath et al., 2022; Shinn et al., 2023; Madaan et al., 2023; Paul et al., 2023), a scheme to prompt LLMs to evaluate their own generation, we use LLMs to judge the correctness of $s^t$ based on $s^{1:t-1}$. Specifically, the evaluation and generation models use the same backend LLM with different prompts, which consist of few-shot exemplars. We follow previous works of CoT (Wei et al., 2022b) or PAL (Gao et al., 2023) to formulate the generation prompts. To construct the in-context exemplars prompt$_\mathcal{C}$ for the self-evaluation LLM $\mathrm{LM}_\mathcal{C}$, we provide stepwise evaluation examples (as question answering with rationales) in each instance. Inspired by Kadavath et al. (2022), we design prompt$_\mathcal{C}$ in the form of multiple-choice questioning (as shown in Figure 2) to better calibrate the model predictions, where we adopt the token-level probability of option A to represent the correctness score as:

$$\mathcal{C}(s^t) = P_{\mathrm{LM}_\mathcal{C}}(\mathsf{A} \mid \mathsf{prompt}_\mathcal{C}, Q, s^{1:t}) \tag{5}$$

---

[3]For ease of notation, we will use $\mathcal{C}(s^t)$ throughout the paper when there is no confusion.

[4]We will denote the LM generation probability by $\mathcal{P}$ throughout the paper for simplification.

[5]In Appendix A.1, we justify the approximation error rate of Eq 4, which computes normalized probability on the subset $\mathcal{S}$ instead of on the entire set.

# 3 Experiments

## 3.1 Setup

We present and analyze the results of our self-evaluation guided beam search with different LLM backbones on various reasoning benchmarks. Implementation details including prompt examples and hyperparameter setup can be found in Appendix A.3.

**Benchmarks.** We evaluate the effectiveness of our approach across three types of reasoning tasks: (1) Arithmetic Reasoning on five math word problem benchmarks, including GSM8K (Cobbe et al., 2021) on math word problems, AQuA (Ling et al., 2017) on algebraic word problems, SVAMP (Patel et al., 2021) on structure variations of math word problems, ASDiv (Miao et al., 2020) on diverse math word problems, and TabMWP (Lu et al., 2023) on tabular math word problems; (2) Symbolic Reasoning on BIG-Bench (Srivastava et al., 2022), which involves `Date Understanding` of context-based date inferring and `Object Counting` of enumerating and counting objects of different types; (3) Commonsense Reasoning on three benchmarks, including CommonsenseQA (Talmor et al., 2019) of commonsense questions that require prior world knowledge to answer, StrategyQA (Geva et al., 2021) of questions that require a multi-hop strategy to answer, and `Sports Understanding` from BIG-Bench (Srivastava et al., 2022) to determine whether a sports-related sentence is plausible.

**Baselines.** We consider two types of baselines: (1) Chain-of-Thought (CoT) (Wei et al., 2022b) prompting in free-text reasoning and (2) Program-Aided Language models (PAL) (Ling et al., 2017) and Program-of-Thought (PoT) (Chen et al., 2022) prompting in program-aided reasoning. We also include their self-consistency (Wang et al., 2023) variants for multiple-chain reasoning. For generation, we follow the few-shot exemplars of baselines. For self-evaluation, we manually create a set of few-shot exemplars based on the baseline outputs on corresponding training data. We formulate self-evaluation as a task of multiple-choice question answering, following Kadavath et al. (2022). For baselines, we represent the cost as the number of generated tokens. For the cost of our method, we also include the number of additional input tokens in self-evaluation for a fair comparison.

**Backboned LLMs.** We assess our approach on closed- and open-source LLMs using both PAL and CoT prompting. For closed-source LLMs, we choose Codex (`code-davinci-002`) (Chen et al., 2021) to report and compare the results on all datasets. We use Llama-2 (`13B`) (Touvron et al., 2023b) as our open-source LLM to conduct cost–performance analysis on different datasets.

## 3.2 Main Results

**Arithmetic and Symbolic Reasoning.** Table 1 shows the results for arithmetic and symbolic reasoning. Our method achieves significant performance improvements on most benchmarks in both single- ($\tau = 0$) and multiple-chain scenarios, with PAL as the baseline. For arithmetic reasoning, we observe absolute increases in accuracy of $5.3\%$, $8.3\%$, and $0.7\%$ on GSM8K, AQuA, and SVAMP, respectively. One possible explanation for this discrepancy in improvements is the reduced diversity in LLM generations due to higher confidence in predictions, as evidenced by the relatively high performance on the tasks. This highlights the importance of incorporating controllable randomness into the candidate generations to expand the search space for self-evaluation guided decoding. We further explore the impact of generation diversity by varying the temperature $\gamma$ in Section 3.4.

For symbolic reasoning, our approach also leads to consistent performance gains. However, when the baseline itself performs well on the task (*e.g.*, $96.7\%$ on `Object Counting`), our approach may not yield substantial improvement. This can also be attributed to the constrained accessible search space for self-evaluation guidance to refine the generation distribution. This limit suggests the inherent deficiency in our LLM-based prompting method that it becomes increasingly challenging to calibrate the generation direction when the model $\text{LM}_{\mathcal{G}}$ is more confident in its predictions. In other words, the high baseline performance usually indicates lower diversity in the LLM generations even with a large temperature $\gamma$, resulting in a limited accessible search space for the model to find a better solution.

**Commonsense Reasoning.** Table 2 compares methods using CoT prompting in commonsense reasoning. Our approach shows consistent performance improvements across several tasks. For example, on StrategyQA, we achieve an accuracy of $77.2\%$ compared with $73.2\%$ of the baseline.

Table 1: Result comparison (accuracy %) on arithmetic and symbolic reasoning tasks. The best result is in **bold** and the lowest cost is in green. We report methods all with Codex backbone for a fair comparison. Similar to Huang et al. (2022), Diverse (Li et al., 2022) fine-tune task-specific verifiers to apply weights on samples in self-consistency (SC). Other fine-tuning methods include reward-based supervision (Uesato et al., 2022) and content-specific training (Lewkowycz et al., 2022). We also report the number of tokens (# Tokens) on GSM8K to compare the costs of different methods.

| Approach | Arithmetic | | | | | | Symbolic | |
| | GSM8K | # Tokens | AQuA | SVAMP | ASDiv | TabMWP | DATE | OBJECT |
| --- | --- | --- | --- | --- | --- | --- | --- | --- |
| *single reasoning chain* | | | | | | | | |
| CoT | 65.6 | 0.2$k$ | 45.3 | 74.8 | 76.9 | 65.2 | 64.8 | 73.0 |
| PoT | 71.6 | — | 54.1 | 85.2 | — | 73.2 | — | — |
| PAL | 72.0 | 0.3$k$ | — | 79.4 | 79.6 | — | 76.2 | 96.7 |
| Ours-PAL | 80.2 | 27.7$k$ | 55.9 | 89.6 | 84.9 | 79.1 | **78.6** | **96.8** |
| *multiple reasoning chains* | | | | | | | | |
| CoT, SC | 78.0 | 5.3$k$ | 52.0 | 86.8 | — | 75.4 | — | — |
| CoT, Diverse | 82.3 | — | — | 87.0 | **88.7** | — | — | — |
| PoT, SC | 80.0 | — | 58.6 | 89.1 | — | **81.8** | — | — |
| PAL, SC | 80.4 | 7.4$k$ | — | — | — | — | — | — |
| Ours-PAL | **85.5** | 550.0$k$ | **64.2** | **90.3** | 85.8 | 80.9 | — | — |

Table 2: Result comparison (accuracy %) on commonsense reasoning tasks, with Codex backbone. Here we only report results in the single reasoning chain scenario following Wei et al. (2022b). We report # Tokens on StrategyQA for cost comparison.

| Approach | StrategyQA | # Tokens | CommonsenseQA | Sports |
| --- | --- | --- | --- | --- |
| CoT | 73.2 | 0.06$k$ | 77.9 | **98.5** |
| Ours-CoT | **77.2** | 11.6$k$ | **78.6** | 98.4 |
| Human | 87.0 | — | 88.9 | — |

Likewise, the performance of our approach is constrained by the low diversity of LLM generations on `Sporting Understanding`, as we observe on `Object Counting` in symbolic reasoning.

**Computational Cost Overhead.** Despite the fact that our approach achieves significant improvement on various benchmarks, we observe an overhead of computational cost compared with the corresponding baselines. For example, the single-chain version of our approach using PAL costs about 3 times more than the self-consistency baseline on GSM8K. As detailed in Appendix A.3, this is due to a relatively large hyperparameter – the number of rollouts per beam $n$ – which we set as 16 for better performance. To strike a balance between performance and cost and present a complete picture, we adopt $n = 2$ and conduct cost–performance analysis on our approach in Section 3.3.

### 3.3 Cost Analysis

Table 3 compares the baseline and our approach under comparable computational budgets (measured in # Tokens). Our method consistently outperforms self-consistency on the arithmetic reasoning tasks even when benchmarked for relatively less computational cost. For example, we achieve 46.1% on GSM8K with a cost of 12.6$k$ tokens, compared with the accuracy of 41.8% of self-consistency which costs 13.9$k$ tokens. Figure 4 further illustrates the cost-efficiency of our approach on GSM8K using different prompting methods under various levels of costs. Our approach significantly outperforms the corresponding equal-cost baseline especially when the computational budget increases, indicating the improvement in the performance upper bound brought by our method.

However, our approach lags behind the CoT baseline on commonsense reasoning. This implies the limitation of our method when applied to shorter reasoning chains, *i.e.*, decreasing the number

Table 3: Cost (# Tokens) and result (accuracy %) comparison on arithmetic and commonsense reasoning tasks. We base our experiments on Llama-2 (13B) since Codex is not available. We show the results of the baseline and our method both in the multiple-chain scenario for a fair comparison. Here we use PAL and CoT prompting for arithmetic and commonsense reasoning, respectively.

| Approach | Arithmetic (PAL) | | | | | Commonsense (CoT) | |
|---|---|---|---|---|---|---|---|
| | GSM8K | AQuA | SVAMP | ASDiv | TabMWP | StrategyQA | CommonsenseQA |
| Baseline | 41.8 | 30.7 | 71.2 | 66.2 | 43.7 | **71.0** | **74.4** |
| # Tokens | 13.9$k$ | 6.6$k$ | 5.9$k$ | 2.7$k$ | 1.9$k$ | 2.7$k$ | 1.2$k$ |
| Ours | **46.1** | **31.5** | **74.6** | **67.7** | **49.6** | 70.6 | 74.0 |
| # Tokens | 12.6$k$ | 6.0$k$ | 5.0$k$ | 2.5$k$ | 1.2$k$ | 2.6$k$ | 1.2$k$ |

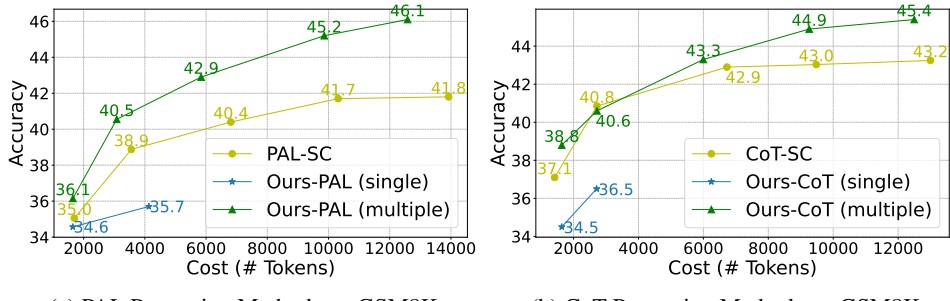

(a) PAL Prompting Methods on GSM8K      (b) CoT Prompting Methods on GSM8K

Figure 4: Accuracy curves on GSM8K of different methods with the change of the cost. We conduct the performance comparison using both PAL and CoT prompting with Llama-2 (13B) backbone.

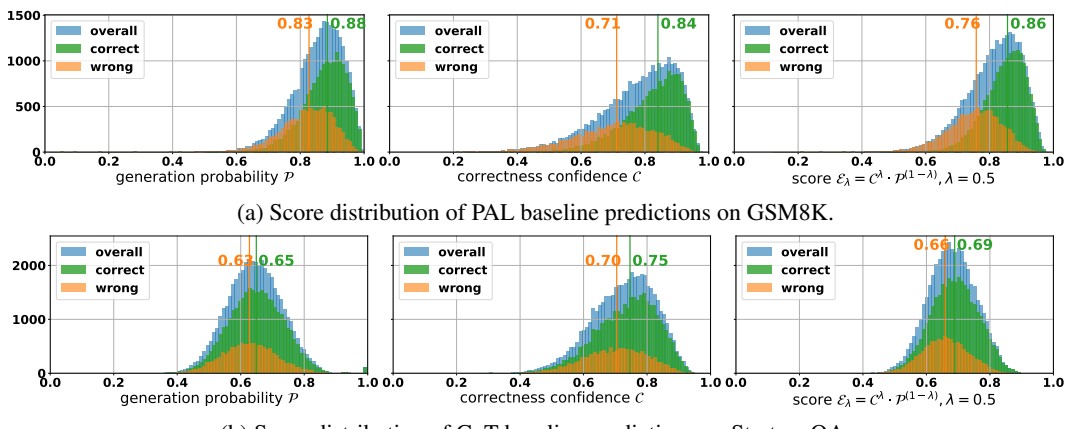

(a) Score distribution of PAL baseline predictions on GSM8K.

(b) Score distribution of CoT baseline predictions on StrategyQA.

Figure 5: Distributions of the self-evaluation score and its components (*i.e.*, generation confidence $\mathcal{P}$ and correctness confidence $\mathcal{C}$) on correct/incorrect baseline predictions. We highlight the median scores of the positive and negative cases using lines of the same colors respectively.

of intermediate steps weakens the effect of stepwise self-evaluation in beam search in reducing error accumulation. On the other hand, self-consistency can directly improve performance through instance-level aggregation without additional cost for self-evaluation. We analyze how our method benefits longer reasoning chains on different tasks in Section 3.4.

## 3.4 Further Analysis

We now provide a detailed analysis of why our method achieves significant gains.

**Generation and Self-evaluation Calibration.** We investigate the distributions of generation confidence (*i.e.*, the LM probability $\mathcal{P}$) and correctness confidence $\mathcal{C}$ in our self-evaluation score $\mathcal{E}$. By

Table 4: Absolute accuracy (in %) increases on instances of different complexity determined by the length of reasoning chains (represented as # Steps).

| | GSM8K | | | | | StrategyQA | | | |
|---|---|---|---|---|---|---|---|---|---|
| # Steps | # Ins. | PAL | Ours | ΔAccu. | # Steps | # Ins. | CoT | Ours | ΔAccu. |
| $< 7$ | 437 | 85.8 | 91.3 | +5.49 | $< 4$ | 637 | 84.6 | 84.9 | +0.31 |
| $\in (7, 9]$ | 524 | 74.8 | 82.6 | +7.82 | $\in [4, 5)$ | 1,301 | 78.6 | 79.1 | +0.46 |
| $\geq 9$ | 358 | 72.9 | 82.6 | +9.78 | $\geq 5$ | 351 | 68.4 | 71.8 | +3.42 |

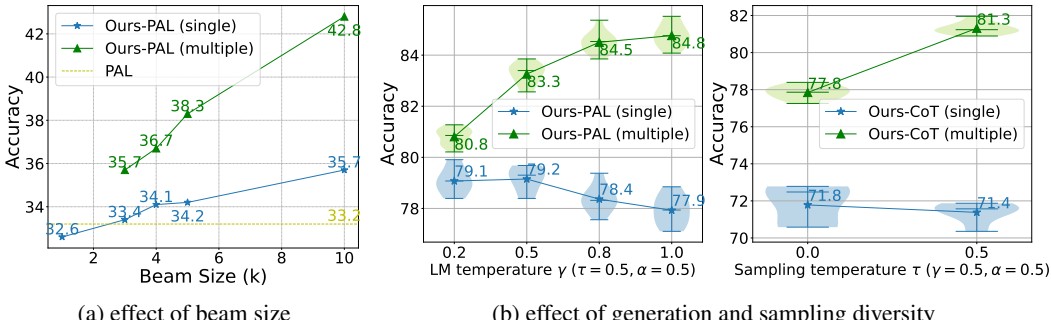

(a) effect of beam size      (b) effect of generation and sampling diversity

Figure 6: Accuracy curves and distributions of our approach on GSM8K with different hyperparameter settings: (a) Changes in performance (Llama-2 backboned) when the beam size $k$ varies. Methods of the same $k$ have equal computational costs; (b) Accuracy distributions (Codex backboned) with different generation temperature $\gamma$ and sampling temperature $\tau$ (with decay ratio $\alpha$).

comparing the score distributions for correct and wrong predictions, we aim to gain an intuitive understanding of whether these confidence scores are reliable. Figure 5 shows different score distributions on correct and wrong baseline predictions. The difference in distribution between the two prediction sets is substantial for arithmetic reasoning, but negligible for commonsense reasoning. Notably, in both instances, correctness confidence is more discriminatory than generation confidence.

To achieve a balance between these two confidence scores, we utilize a tunable hyperparameter $\lambda$, setting $\lambda = 0.5$ for all datasets. Nevertheless, varying its value can lead to distinct outcomes. For instance, when setting $\lambda$ to 1 ($\mathcal{E} = \mathcal{C}$) or 0 ($\mathcal{E} = \mathcal{P}$), the performance on GSM8K decreases from 80.2% to 74.5% and 77.1%, respectively. This indicates that both scores play a crucial role in our final performance. A more comprehensive analysis of $\lambda$ can be found in Appendix A.2.

**Reasoning Complexity.** We investigate if our approach is more beneficial for instances needing more reasoning steps. Table 4 shows that performance gains (in absolute accuracy % increase) increase as reasoning chains become longer on both GSM8K and StrategyQA. Notably, the improvement on StrategyQA primarily comes from improvements in longer reasoning chains, showcasing the effectiveness of our method in navigating lengthy and intricate reasoning chains.

**Hyperparameters in Stochastic Beam Search.** We examine the significance of hyperparameters associated with stochastic beam search, including the beam size $k$ and the temperatures $\gamma$ and $\tau$ controlling the generation and sampling diversity, respectively.

Figure 6a shows the trend of performance improvement with the increase of beam size $k$. Notably, our beam search approach inherently enables majority voting on the final beam without additional cost, resulting in a more significant performance improvement in the multiple-chain reasoning when the beam size is larger (e.g., 42.8% compared with 35.7% when $k = 10$).

For generation and sampling diversity, it is clear that more diversity resulting from higher temperatures generally leads to a decline in performance when only considering a single reasoning chain. However, diversity significantly benefits majority voting on multiple reasoning chains [6]. This benefit comes

---

[6]In this study, we did not explore higher generation temperatures (i.e., $\gamma > 1.0$) since this hyperparameter is limited to 1.0 in the OpenAI API.

(a) Examples of self-evaluation score distribution of different predictions on the GSM8K dataset.

(b) Examples of self-evaluation score distribution of different predictions on the StrategyQA dataset. We also provide explanations corresponding to the ground-truth answers for reference.

Figure 7: Comparisons among predictions of high and low self-evaluation scores on arithmetic (7a for GSM8K) and commonsense (7b for StrategyQA) reasoning tasks. Scores from low to high are visualized from orange (0.0), yellow (0.4), to green (1.0). Here $\mathcal{C}, \mathcal{P}$, and $\mathcal{E}$ represent the evaluation confidence, the generation confidence, and their combination as the final score, respectively.

from the improved coverage of the plausible generations and the ensembling effect. Nevertheless, one can adjust the sampling-related parameters (*i.e.*, $\tau$ and $\alpha$) to incorporate more randomness into the generations. In practice, we find that a moderate temperature decay (*e.g.*, $\alpha = 0.5$) results in improved performance. We conduct further analysis of the effect of sampling diversity in Appendix A.2.

**Qualitative Analysis.** We examine particular instances to investigate the behavior of correctness confidence scores $\mathcal{C}$ and generation probabilities $\mathcal{P}$ in different scenarios. From the comparison shown in Figure 7, we have the following main observations:

• In general, the correctness confidence is more effective at identifying logical errors, taking into account the accumulated mistakes from prior steps, while the generation probability focuses more on text perplexity as the confidence of the generation LLM.

• When comparing arithmetic and commonsense tasks, LLMs exhibit greater confidence in dealing with structured and objective reasoning chains such as problems in GSM8K, for both generation and self-evaluation, as opposed to reasoning chains in StrategyQA.

• Reasoning chains that appear logically plausible can achieve high correctness confidence scores but still result in incorrect answers, as demonstrated in $R_{41}$ in Figure 7b. Moreover, the correctness confidence can be influenced by minor details (*e.g.*, imperfect variable naming in PAL reasoning) and assign low scores regardless of the correctness of the final answers as shown in $R_{22}$ in Figure 7a.

• Incoherence due to a sudden jump in reasoning (*e.g.*, $R_{32}$ in Figure 7b) can lead to low correctness confidence. Additionally, the correctness confidence tends to be lower when the generation LLM makes a probability statement with less certainty, such as "it seems" as illustrated by $R_{42}$ in Figure 7b.

# 4 Related Work

**Reasoning Formulation.**  Several studies have attempted to better formulate the reasoning problem. One approach is to generate rationales to enhance model interpretability (Zhou et al., 2020; Wiegreffe and Marasovic, 2021; Wiegreffe et al., 2021). Recently, the focus has shifted towards decomposing the reasoning process into intermediate steps before reaching the final answer (Wei et al., 2022b; Zhou et al., 2023; Gao et al., 2023; Chen et al., 2022). Various decomposition techniques have been explored, such as question reduction (Zhou et al., 2023; Yang et al., 2022), iterative prompting (Wang et al., 2022), and chaining the steps (Wu et al., 2022). While incorporating intermediate reasoning steps has resulted in substantial performance improvements, errors or imperfections can accumulate, especially when the chains become longer (Wu et al., 2016; Guo et al., 2018). As such, we utilize LLM self-evaluation as a stepwise criterion to improve the chaining process.

**LLM Self-Evaluation.**  Recent research on LLM calibration shows that current LLMs' probabilistic predictions correspond well with actual token occurrence frequencies, leading to well-calibrated predictions for specific tasks (Rae et al., 2021; Kadavath et al., 2022; Guo et al., 2017; Kadavath et al., 2022; Jiang et al., 2021; Kuhn et al., 2023). Notably, scaling model size plays a crucial role in enhancing calibration (Rae et al., 2021; Wei et al., 2022a). As LLMs exhibit good calibration, an increasing number of studies focus on prompting LLMs to perform self-evaluation as a means of verification (Zhang et al., 2023; Shinn et al., 2023; Madaan et al., 2023; Paul et al., 2023). Self-evaluation provides an effective and efficient assessment method without requiring task-specific verifier fine-tuning, which typically involves additional annotations (Li et al., 2022). In contrast to existing works that refine generation results through instance-level self-evaluation, our approach applies self-evaluation results as a stepwise criterion to calibrate generation at a finer granularity. By focusing on step-by-step self-evaluation, our method enables fine-grained guided decoding, addressing the challenges associated with complex or lengthy reasoning.

**Decoding Strategies.**  A tradeoff typically exists between diversity and quality. Deterministic decoding methods such as greedy decoding and beam search (Jurafsky and Martin, 2009; Graves, 2012) often produce high-quality results but lack diversity (Stahlberg and Byrne, 2019; Meister et al., 2020). Temperature sampling (Ackley et al., 1985), top-$k$ sampling (Fan et al., 2018), and top-$p$ sampling (Holtzman et al., 2020) are various techniques used to enhance diversity. The recent work of *tree-of-thought* (Yao et al., 2023) explores different search algorithms such as breadth-first and depth-first searches tailored for different tasks. Differently, we propose a unified framework of stochastic beam search (Caccia et al., 2020; Kool et al., 2019; Meister et al., 2021), which combines beam search and temperature sampling to balance the quality–diversity trade-off in multi-step reasoning.

# 5 Discussion

We have introduced a multi-step decoding method that calibrates reasoning with stepwise self-evaluation guidance via stochastic beam search for current large language models. The empirical success of our method across a broad range of tasks, from arithmetic and symbolic to commonsense reasoning, demonstrates its robustness and generalizability in various application areas. The significant performance gains of our method on long reasoning chains also highlight its applicability to other multi-step tasks, such as multi-hop question answering and more complex scenarios involving multi-modal understanding, reasoning, and planning. In future work, we will investigate how to utilize external tools to further enhance the calibration and explore its generalizability on other multi-step scenarios to deal with more complex information such as external knowledge and multimodalities.

## Potential Impacts and Limitations

We propose self-evaluation guided stochastic beam search to facilitate multi-step reasoning. However, our approach, based on stepwise self-evaluation guidance, has certain limitations. It requires access to LLM logits to calculate the self-evaluation score, restricting its applicability to more powerful LLMs, such as GPT-4, which do not provide token likelihoods. Plus, multi-step decoding inherently causes additional costs from candidate sampling and self-evaluation. For optimal balance between efficiency and cost, our approach is best applied to longer reasoning chains, where the cumulative effect of calibration across multiple steps can improve the overall performance more significantly.

## Acknowledgments and Disclosure of Funding

The computational work for this article was partially performed on resources of the National Super-computing Centre (NSCC), Singapore[7]. We would like to thank Prof. Hwee Tou Ng for his insightful discussions that enhanced the depth and quality of our study.

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

# A  Appendix

## A.1  Theoretical Analysis of Eq. 4

In Eq. 4, we use $\mathcal{S}$ sampled from the language model $\text{LM}_{\mathcal{G}}$ generations. This is an approximation for sampling from the infinite set of all possible chaining paths. And the finite set $\mathcal{S}$ is constructed based on the generation LM $P_{\text{LM}_{\mathcal{G}}}$, which is different from our target distribution as shown in Eq. 4.

Specifically, denote the infinite set of all possible generated completions till the $t$-th step as $\mathcal{S}^*$, we approximate sampling from $P_{beam}^*(s^{1:t}) = \frac{\exp\left(\mathcal{E}(s^{1:t})/\tau\right)}{\sum_{s^{1:t}\in\mathcal{S}^*}\exp\left(\mathcal{E}(s^{1:t})/\tau\right)}$ via $P_{beam}(s^{1:t}) = \frac{\exp\left(\mathcal{E}(s^{1:t})/\tau\right)}{\sum_{s^{1:t}\in\mathcal{S}}\exp\left(\mathcal{E}(s^{1:t})/\tau\right)}$, where $\mathcal{S}$ is the approximation of $\mathcal{S}^*$ with $|\mathcal{S}| = nk = M \leq |\mathcal{S}^*|$.

Define the upper bound $\bar{c}$ and the lower bound $\underline{c}$ on each $\exp\left(\mathcal{E}(s^{1:t})/\tau\right)$ as $\bar{c} \geq \exp\left(\mathcal{E}(s^{1:t})/\tau\right) \geq \underline{c}$ for all $s^{1:t} \in \mathcal{S}^*$. Define the ratio as $r = \bar{c}/\underline{c}$. Note that $\underline{c} \geq 1$ since $\mathcal{E}(s^{1:t})/\tau \geq 0$. Thus, we can take $r \leq \bar{c}$.

We now give the following proposition which shows that $|P_{beam}^*(s^{1:t}) - P_{beam}(s^{1:t})|$ decreases at the rate of $\mathcal{O}(\frac{1-M/|\mathcal{S}^*|}{M})$ toward 0 as $M$ increases. Note that as $M$ increases toward $|\mathcal{S}^*|$, the numerator $1 - M/|\mathcal{S}^*|$ decreases toward 0 while the factor for the denominator $\frac{1}{M}$ also decreases.

**Proposition 1.** *For any $s^{1:t}$, the difference between $P_{beam}^*(s^{1:t})$ and $P_{beam}(s^{1:t})$ is bounded by*

$$|P_{beam}^*(s^{1:t}) - P_{beam}(s^{1:t})| \leq r^2 \left(\frac{1 - M/|\mathcal{S}^*|}{M}\right) \tag{6}$$

*Proof.*  We now prove the second statement by analyzing the absolute difference:

$$|P_{beam}^*(s^{1:t}) - P_{beam}(s^{1:t})| \tag{7}$$

$$= \left| \frac{\exp\left(\mathcal{E}(s^{1:t})/\tau\right)}{\sum_{s^{1:t}\in\mathcal{S}^*}\exp\left(\mathcal{E}(s^{1:t})/\tau\right)} - \frac{\exp\left(\mathcal{E}(s^{1:t})/\tau\right)}{\sum_{s^{1:t}\in\mathcal{S}}\exp\left(\mathcal{E}(s^{1:t})/\tau\right)} \right| \tag{8}$$

$$= \frac{\exp\left(\mathcal{E}(s^{1:t})/\tau\right)\left|\sum_{s^{1:t}\in\mathcal{S}^*}\exp\left(\mathcal{E}(s^{1:t})/\tau\right) - \sum_{s^{1:t}\in\mathcal{S}}\exp\left(\mathcal{E}(s^{1:t})/\tau\right)\right|}{\left(\sum_{s^{1:t}\in\mathcal{S}}\exp\left(\mathcal{E}(s^{1:t})/\tau\right)\sum_{s^{1:t}\in\mathcal{S}^*}\exp\left(\mathcal{E}(s^{1:t})/\tau\right)\right)} \tag{9}$$

$$= \frac{\exp\left(\mathcal{E}(s^{1:t})/\tau\right)\left|\sum_{s^{1:t}\in\mathcal{S}^*\backslash\mathcal{S}}\exp\left(\mathcal{E}(s^{1:t})/\tau\right)\right|}{\left(\sum_{s^{1:t}\in\mathcal{S}}\exp\left(\mathcal{E}(s^{1:t})/\tau\right)\right)\sum_{s^{1:t}\in\mathcal{S}^*}\exp\left(\mathcal{E}(s^{1:t})/\tau\right)} \tag{10}$$

Since $\exp\left(\mathcal{E}(s^{1:t})/\tau\right)$ is nonnegative, using the upper bound on each $\exp\left(\mathcal{E}(s^{1:t})/\tau\right)$, we have:

$$|P_{beam}^*(s^{1:t}) - P_{beam}(s^{1:t})| \leq \frac{\bar{c}^2(|\mathcal{S}^*| - M)}{\left(\sum_{s^{1:t}\in\mathcal{S}}\exp\left(\mathcal{E}(s^{1:t})/\tau\right)\right)\sum_{s^{1:t}\in\mathcal{S}^*}\exp\left(\mathcal{E}(s^{1:t})/\tau\right)} \tag{11}$$

Similarly, using the lower bound on each $\exp\left(\mathcal{E}(s^{1:t})/\tau\right)$,

$$|P_{beam}^*(s^{1:t}) - P_{beam}(s^{1:t})| \leq \frac{\bar{c}^2(|\mathcal{S}^*| - M)}{\underline{c}^2|\mathcal{S}^*|M} = r^2 \left(\frac{1 - M/|\mathcal{S}^*|}{M}\right) \tag{12}$$

$\square$

## A.2    Extended Experiments

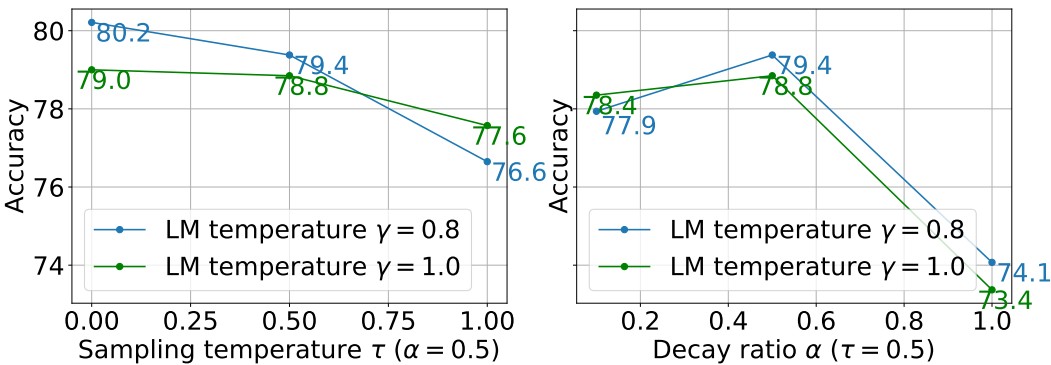

Figure 8: Accuracy curves with different sampling diversity. The two plots show the changes in performance on GSM8K when the sampling temperature $\tau$ and its decay ratio $\alpha$ vary, respectively.

**Sampling Diversity.**    In accordance with Figure 6b, we observe similar results when ablating the sampling hyperparameters $\tau$ and $\alpha$ for the single reasoning chain case, as shown in Figure 8. Increasing $\tau$ and $\alpha$ generally adds more diversity to the decoding process, but excessive randomness negatively impacts the performance of the single-chain decoding. Generally, a moderate temperature decay results in improved performance. Therefore, we set $\alpha = 0.5$ throughout our experiments for simplicity and only tune $\tau$ for randomness control.

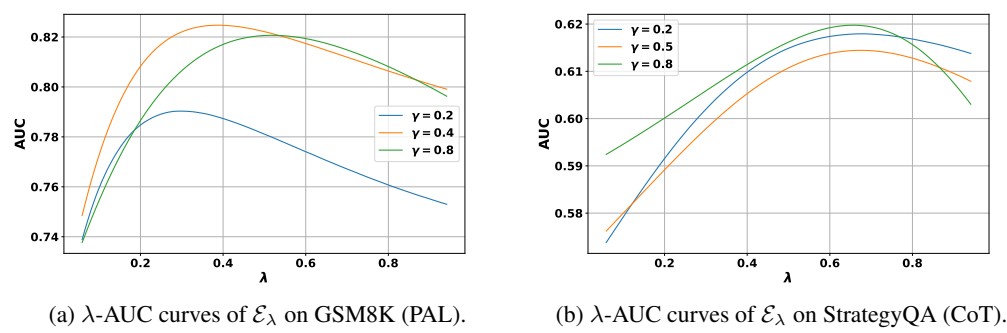

(a) $\lambda$-AUC curves of $\mathcal{E}_\lambda$ on GSM8K (PAL).    (b) $\lambda$-AUC curves of $\mathcal{E}_\lambda$ on StrategyQA (CoT).

Figure 9: The change of AUC scores with different values of $\lambda$ in $\mathcal{E}_\lambda$. We calculate the AUC score as how $\mathcal{E}_\lambda$ can successfully determine whether the corresponding predicted reasoning chain can produce the ground-truth answer. The predictions here are from the baseline methods (*i.e.*, CoT & PAL) with different LM temperatures $\gamma$, as represented by curves of different colors.

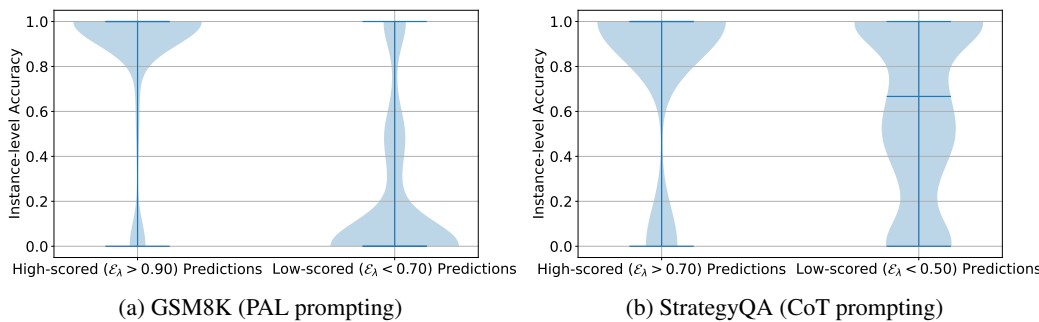

(a) GSM8K (PAL prompting)    (b) StrategyQA (CoT prompting)

Figure 10: Comparison between predictions of high v.s. low self-evaluation scores on instance-level accuracy.

**More Analysis on Self-Evaluation.** Recall that we use a combination of generation confidence and faithfulness score as $\mathcal{E}_\lambda = \mathcal{C}^\lambda \cdot \mathcal{P}^{(1-\lambda)}$, with $\lambda \in [0, 1]$. In our experiments, we set $\lambda = 0.5$ for all tasks for simplicity. However, we investigate its effects here since, intuitively, it is an important hyperparameter for distinguishing correct / incorrect predictions and might require different values for various reasoning tasks and datasets. Its effect is also coupled with the language model temperature $\gamma$.

Figure 9 demonstrates how $\lambda$ functions on arithmetic (GSM8K) and commonsense (StrategyQA). In general, we observe that the performance remains relatively stable with different choices of $\lambda$ on different datasets, although fine-tuning this hyperparameter might lead to further improvements. This stability suggests that the choice of $\lambda$ is not overly sensitive across various reasoning tasks and datasets, but exploring its optimal value for specific tasks could potentially lead to even better performances.

To examine the influence of incorporating faithfulness on LLM final predictions, we plot the distributions of changes in different scores, specifically the faithfulness score $\mathcal{C}$, the generation confidence $\mathcal{P}$, and the overall decoding score $\mathcal{E}_\lambda$ on the baseline reasoning chains and the reasoning chains generated by our method. We categorize the data points into 4 sets based on whether our approach changes the final prediction. Since the majority of the data points belong to the "both correct" set (in blue), where both baselines and our method generate accurate predictions, we particularly highlight the last two sets (in green and red), where our method results in improvement and degradation, respectively.

As shown in Figure 11, faithfulness typically works by significantly increasing the evaluation confidence $\mathcal{C}$ of model predictions, while the generation confidence $\mathcal{P}$ remains similar to that of the baseline methods. Specifically, for the evaluation confidence $\mathcal{C}$, our approach corrects the original predictions by increasing the confidence scores. This indicates that evaluation confidence plays a crucial role in guiding the decoding toward a better reasoning choice in decomposed reasoning. The increase is more significant for PAL when compared with CoT. This demonstrates that LLMs are generally better at judging the logic in reasoning that is more structured, while the free-text intermediate steps (*e.g.*, CoT reasoning) may be challenging to conduct information extraction and soundness checking.

A similar conclusion can be drawn from Figure 10, where the difference in instance-level accuracy distributions between high-scored and low-scored predictions is more significant on the GSM8K dataset. For StrategyQA, while the incorporation of faithfulness helps, the level of the score value does not align well with whether the prediction is correct. For example, most of the low-scored predictions can still obtain the correct answers, as shown by the plot on the right of Figure 10b.

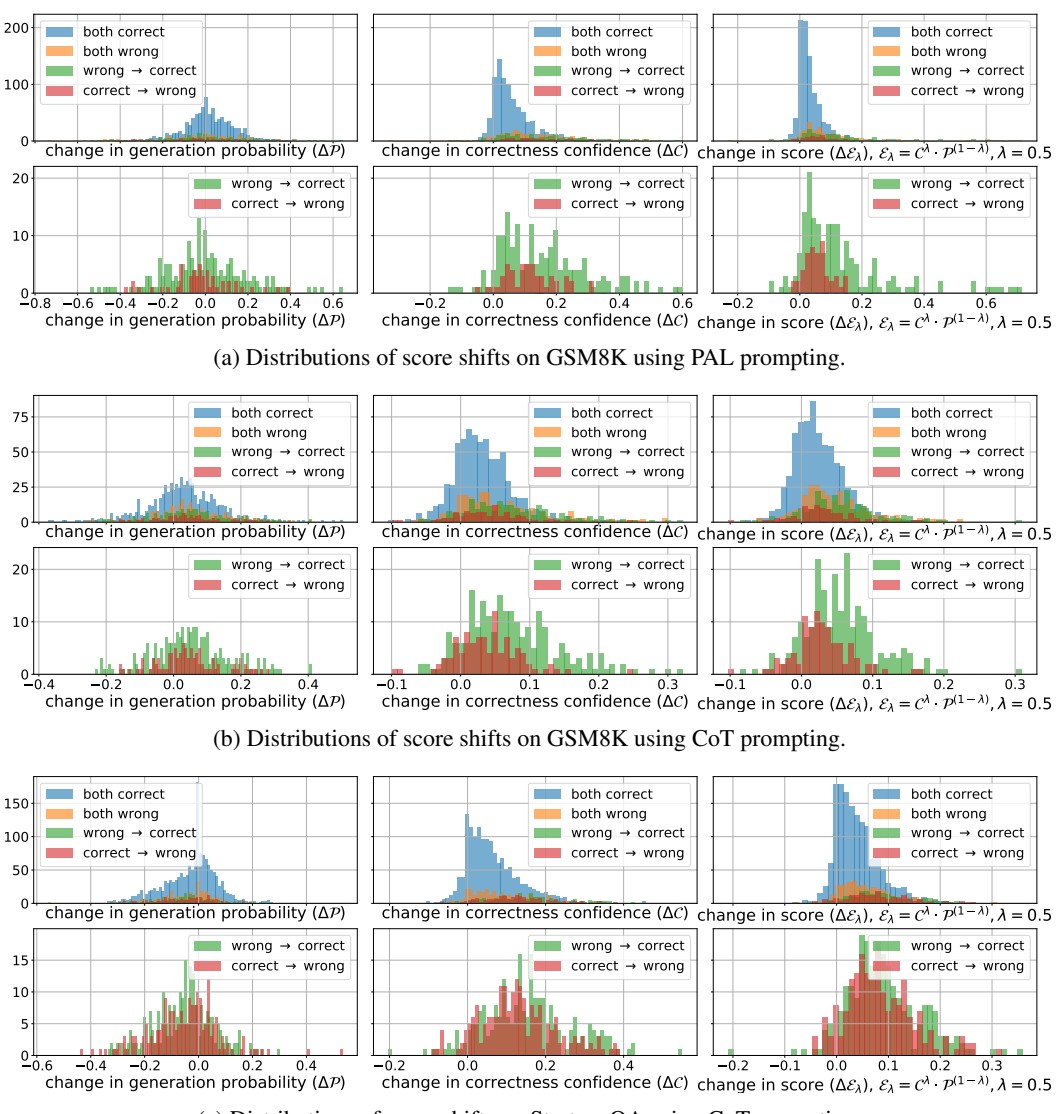

(a) Distributions of score shifts on GSM8K using PAL prompting.

(b) Distributions of score shifts on GSM8K using CoT prompting.

(c) Distributions of score shifts on StrategyQA using CoT prompting.

Figure 11: Distributions of changes in scores from baselines to our method. Since the prediction correctness keeps unchanged most of the time (*i.e.*, "both correct/incorrect" in blue/orange), we specifically plot how the scores shift on data points where the predictions get corrected or incorrect, as shown in green and red, respectively.

Table 5: Impact of LLM backends (Codex vs. ChatGPT vs. GPT-4) and prompting methods (PAL vs. CoT). The results of ChatGPT (`gpt-3.5-turbo`) were obtained on 20 March 2023.

| Model | GSM8K | StrategyQA |
|---|---|---|
| CoT $_{\text{Codex}}$ | 65.6 | 73.2 |
| PAL $_{\text{Codex}}$ | 72.0 | — |
| CoT $_{\text{ChatGPT}}$ | 80.8 | 65.9 |
| PAL $_{\text{ChatGPT}}$ | 78.7 | — |
| CoT $_{\text{GPT}-4}$ | **92.0** | — |
| Ours (CoT) $_{\text{Codex}}$ | 71.9 | **77.2** |
| Ours (PAL) $_{\text{Codex}}$ | 80.2 | — |

**LLM Backbone Study.** We are interested in how stronger LLMs (*i.e.*, ChatGPT, GPT-4 (OpenAI, 2023)) work, but they are not directly compatible with our approach since the API does not return token logits.

Table 5 compares the results of various backend LLMs (*i.e.*, Codex, ChatGPT, and GPT-4) on GSM8K. In arithmetic reasoning with PAL prompting, our Codex-based method achieves competitive results (80.2% vs. 78.7%) even when compared with ChatGPT. The results are consistent across other datasets, including AQuA (55.9% vs. 54.7%), SVAMP (89.6% vs. 84.1%), ASDiv (84.9% vs. 84.1%), and TabMWP (79.1% vs. 80.6%). In commonsense reasoning, our method using Codex significantly outperforms ChatGPT-based methods across different datasets, including StrategyQA (77.2% vs. 65.9%), CommonsenseQA (78.6% vs. 75.2%) and `Sports Understanding` (98.4% vs. 95.9%). One possible explanation is that ChatGPT lacks sufficient world knowledge for effective fact checking and commonsense reasoning. Given the significant performance improvement of GPT-4, we conduct further analysis about how to synergistically combine it with our method.

**GPT-4 Experiments** The recently launched GPT-4 has demonstrated notable improvements in reasoning capabilities across a variety of tasks. In this section, we examine and compare the reasoning skills of different large language models (LLMs), specifically Codex and GPT-4, in assessing and determining the accuracy of each step in a reasoning chain. We contrast the confidence scores and corresponding explanations for Codex ($\mathcal{C}$) and GPT-4 ($\mathcal{S}$) in the context of both arithmetic and commonsense reasoning, as shown in Figure 13 and Figure 14, respectively. For ease of visualization, we employ the same colormap (shown in Figure 12) as in Figure 7 to represent the scale of scores. Since OpenAI has not provided access to the token-wise likelihood of generated text, we directly request GPT-4 to score the reasoning steps using binary values [8]. Moreover, we report the average of three evaluation results to reduce the variance of sampling discrete values, *i.e.*, $S = (S_1 + S_2 + S_3)/3, S_i \in [0, 1]$.

As illustrated in Figure 13, GPT-4 demonstrates greater effectiveness in pinpointing the central logical error in arithmetic reasoning. For instance, we can observe that $\mathcal{S} < \mathcal{C}$ for `alex_total = alex_weight + weight_multiplier * grace_weight` and $\mathcal{S} > \mathcal{C}$ for `answer = grace_weight + alex_total`, where the former leads to an incorrect final answer. Additionally, GPT-4 typically offers detailed explanations and alternative solutions. As seen in the step `answer = grace_weight + alex_total`, GPT-4 can correct minor errors even when it arrives at the correct final answer. However, GPT-4 may still encounter difficulties in detecting small errors within the text, which can have a significant impact on logical consistency. This challenge is illustrated by the substantial variance in $\mathcal{S}$ for the step `alex_total = alex_weight + weight_multiplier * grace_weight`.

The benefits of well-crafted explanations in GPT-4 become more significant when handling complex reasoning tasks, as demonstrated in Figure 14. For instance, in the $R_{42}$ of $Q_4$ shown in Figure 7b, Codex generally assigns high evaluation scores for all steps. Although this reasoning chain leads to the correct final answer, it makes some overly definitive assumptions without reasonable justification (e.g., "must have attributes that match both"). In such cases, GPT-4 can accurately identify these vague statements through meticulous analysis. Moreover, the comprehensive analysis helps address the growing uncertainty arising from the ambiguity in understanding commonsense questions. For example, in the final step, GPT-4 offers extensive explanations for assigning low $\mathcal{S}$ scores, considering the limited information available to comprehend the question.

In summary, GPT-4 demonstrates promising improvements in LLM evaluation by providing more rigorous and in-depth explanations. While there is significant variation in its generated explanations for some challenging cases, these explanations are typically beneficial as they elaborate on understanding and interpretation based on the given information. This suggests the potential for using LLM explanations to enhance self-evaluation in future research.

---

[8]The results using GPT-4 were obtained on 22 April 2023.

Figure 12: Score from low to high values visualized using colormap from orange to green.

**[$Q_1$] Grace** weighs **125** pounds. **Alex** weighs **2** pounds less than **4** times what **Grace** weighs. What are their **combined** weights in pounds?
**[Ground-Truth $a_1$*]** 623.0

**[Predicted $a_{12}$]** 627 ✗
**[$R_{12}$]** in Python

```python
grace_weight = 125
```

| $\mathcal{E}$ | $\mathcal{C}$ | correct |
|---|---|---|
| $\mathcal{S}$ | $S_1$ | correct |
| | $S_2$ | The line of code correctly assigns the given weight of Grace (**125** pounds) to the variable `grace_weight` |
| | $S_3$ | The given line of code correctly assigns the value of **125** to the variable `grace_weight`, representing Grace's weight in pounds. |

```python
alex_weight = 2
```

| $\mathcal{E}$ | $\mathcal{C}$ | `alex_weight` should be calculated based on `grace_weight` |
|---|---|---|
| $\mathcal{S}$ | $S_1$ | `alex_weight` should be calculated as **4** times `grace_weight` minus **2** |
| | $S_2$ | The line of code defining `alex_weight` should be calculated as **4** times Grace's weight minus **2**, which is (**4** * `grace_weight`) - **2**. |
| | $S_3$ | The calculation for Alex's weight is not correct. It should be calculated as **4** * `grace_weight` - **2**. |

```python
weight_multiplier = 4
```

| $\mathcal{E}$ | $\mathcal{C}$ | correct |
|---|---|---|
| $\mathcal{S}$ | $S_1$ | The line of code is correct as it defines the multiplier for Alex's weight based on Grace's weight. |
| | $S_2$ | Correct because it assigns the correct value (**4**) to the variable `weight_multiplier`, which represents the factor by which Grace's weight is multiplied to calculate Alex's weight. |
| | $S_3$ | The code correctly assigns the value **4** to the variable `weight_multiplier`, which will be used in the calculation of Alex's weight. |

```python
alex_total = alex_weight + weight_multiplier * grace_weight
```

| $\mathcal{E}$ | $\mathcal{C}$ | correct formula to calculate `alex_total` |
|---|---|---|
| $\mathcal{S}$ | $S_1$ | The formula used to calculate Alex's weight is incorrect. |
| | $S_2$ | The above line of code calculates Alex's weight correctly as **2** pounds less than **4** times Grace's weight (**4** * `grace_weight` - **2**). |
| | $S_3$ | Alex weighs **2** pounds less than **4** times what Grace weighs, not **2** pounds more. |

```python
answer = grace_weight + alex_total
```

| $\mathcal{E}$ | $\mathcal{C}$ | should be the sum of `grace_weight` and `alex_weight` |
|---|---|---|
| $\mathcal{S}$ | $S_1$ | The code calculates Alex's weight correctly using the given relationship and adds it to Grace's weight to calculate the combined weight. The only minor issue is the unnecessary use of the separate variable `alex_weight` which could directly be replaced with the calculation (**4** * `grace_weight` - **2**). However, this does not affect the correctness of the answer. |
| | $S_2$ | The result variable holds the correct combined weight of Grace and Alex, and this line of code correctly calculates the value of `answer`. |
| | $S_3$ | The code above correctly calculates the total weight of Grace and Alex based on the given problem statement. The only minor issue is that the `alex_weight` variable is initialized as **2**, which might be a bit confusing. It would be better to directly calculate `alex_total` as (**4** * `grace_weight` - **2**), but the current code does not produce an incorrect result. |

Figure 13: Comparison between Codex ($\mathcal{C}$) and GPT-4 ($\mathcal{S}$) on evaluating the reasoning chains in arithmetic reasoning. The example ($R_{12}$ for $Q_1$) is the same one from Figure 7a. We underline the key statements in explanations that justify the judgment of GPT-4.

[$Q_4$] Is **Freya** a combination of **Athena** and **Aphrodite**?

[Ground-Truth $a_4$*] yes

[Explanation] **Athena** was the **Greek** goddess of **war**. **Aphrodite** was the **Greek** goddess of **love**. **Freya** was the **Norse** goddess of **war**, **love**, and **fertility**.

[Predicted $a_{42}$] yes ✓

[$R_{42}$] in `Chain-of-Thought`

**Freya is the goddess of love, beauty and fertility.**

| $\mathcal{E}$ | $\mathcal{C}$ | correct |
|---|---|---|
| $\mathcal{S}$ | $S_1$ | Freya indeed is associated with love, beauty, and fertility in **Norse** mythology. However, it is important to note that while the comparison to Athena and Aphrodite is understandable, it is not a direct or complete equivalence. |
| | $S_2$ | The statement correctly identifies Freya. However, the statement does not mention that Freya is a **Norse** goddess, which could slightly impact the understanding of the comparison to **Greek** goddesses Athena and Aphrodite. |
| | $S_3$ | The statement is mostly correct as Freya is indeed a goddess associated with love, beauty, and fertility. However, it might be worth mentioning that she is a **Norse** goddess, while Athena and Aphrodite are **Greek** goddesses, to provide more context. |

**You can compare these to the Greek gods Athena and Aphrodite.**

| $\mathcal{E}$ | $\mathcal{C}$ | correct |
|---|---|---|
| $\mathcal{S}$ | $S_1$ | It is true that Freya can be compared to the **Greek** gods Athena and Aphrodite in terms of their respective domains. However, the comparison is not entirely accurate as Freya comes from **Norse** mythology, and the attributes of the gods may not be an exact match. The comparison is a simplification, which is why the probability is not higher. |
| | $S_2$ | The statement establishes a comparison between Freya and the **Greek** goddesses Athena and Aphrodite. While this step of reasoning is generally correct, it could be improved by specifically mentioning that Freya is a **Norse** goddess and by providing more information about the roles and attributes of Athena and Aphrodite for a clearer comparison. |
| | $S_3$ | While it is true that Freya shares some similarities with Athena and Aphrodite, the comparison is not entirely accurate. Freya is a **Norse** goddess, while Athena and Aphrodite are **Greek** goddesses. The comparison is correct in the sense that they share some domains, but it might be misleading to suggest that Freya is a direct "combination" of these two goddesses. |

**It is not clear how their powers exactly compare.**

| $\mathcal{E}$ | $\mathcal{C}$ | correct |
|---|---|---|
| $\mathcal{S}$ | $S_1$ | This step of reasoning acknowledges that it is not entirely clear how the powers of Freya, Athena, and Aphrodite compare. Although there are some similarities, the differences between **Norse** and **Greek** mythologies make it difficult to determine whether Freya is a direct combination of Athena and Aphrodite. The high probability reflects the acknowledgment of the uncertainty in comparing the powers of the gods across different mythologies. |
| | $S_2$ | The statement acknowledges that it is not clear how the powers of Freya, Athena, and Aphrodite exactly compare. However, the question asks if Freya is a combination of Athena and Aphrodite, which implies a synthesis or blending of their roles or attributes. The reasoning could be improved by specifically mentioning the roles and attributes of Athena and Aphrodite, and whether or not they overlap with Freya's roles and attributes. |
| | $S_3$ | The reasoning here is more accurate, acknowledging that while there are similarities between Freya and Athena and Aphrodite, it is not clear how their powers exactly compare. It is important to consider that Freya is a **Norse** goddess and Athena and Aphrodite are **Greek** goddesses, so their roles, domains, and characteristics may not be directly comparable. |

**However, it seems that Freya must have attributes that match both.**

| $\mathcal{E}$ | $\mathcal{C}$ | correct |
|---|---|---|
| $\mathcal{S}$ | $S_1$ | incorrect because it assumes that Freya must have attributes that match both Athena and Aphrodite. While there may be some similarities in their attributes, it is not appropriate to conclude that Freya is a combination of the two **Greek** goddesses. Norse and **Greek** mythologies are distinct, and the deities in these pantheons may have different origins and roles. |
| | $S_2$ | The statement acknowledges that Freya has attributes that match both Athena and Aphrodite. However, it would be more accurate to discuss the specific attributes of Athena and Aphrodite to determine the extent of the similarities. |
| | $S_3$ | It assumes that Freya must have attributes that match both Athena and Aphrodite, but this may not be the case. While there are similarities between these goddesses, they are from different mythologies (Freya – **Norse**, Athena & Aphrodite – **Greek**), and their roles, domains, and characteristics may not be directly comparable or combined. It is not accurate to assume that Freya is a combination of Athena and Aphrodite without a more in-depth analysis in their respective mythologies. |

**So the answer is yes.**

| $\mathcal{E}$ | $\mathcal{C}$ | correct |
|---|---|---|
| $\mathcal{S}$ | $S_1$ | This step of reasoning is incorrect because it is just based on some similarities in their attributes. While there may be some similarities, it is not accurate to say that Freya is a combination of the two **Greek** goddesses. Norse and **Greek** mythologies are distinct, and the deities in these pantheons have different origins and roles. |
| | $S_2$ | Although Freya shares some attributes with both Athena and Aphrodite, it is an oversimplification to claim that she is a combination of the two. The question seems to imply that Freya might be a direct mix of the two, but this is not accurate. |
| | $S_3$ | The conclusion that Freya is a combination of Athena and Aphrodite is not accurate based on the information provided. While there are similarities between these goddesses, they belong to different mythologies (Freya is **Norse**, Athena and Aphrodite are **Greek**), and their roles, domains, and characteristics may not be directly comparable or combined. |

Figure 14: Comparison between Codex ($\mathcal{C}$) and GPT-4 ($\mathcal{S}$) on evaluating the reasoning chains in commonsense reasoning. The example ($R_{42}$ for $Q_4$) is the same one from Figure 7b. We underline the key points in GPT-4 rationales that explain the detailed understanding and analysis on the steps.

## A.3 Implementation Details

Similar to beam search, we maintain $k$ distinct candidates in the beam and sample $n$ completions for each one. Thus, for each reasoning step $s^t$, the search space has a size of $k \cdot n$. After acquiring $k \cdot n$ samples, we retain $k$ candidates by sampling from $P_{beam}$ as Eq. 4. We set $k = 5, n = 16$ with Codex backbone to balance the quality and efficiency. The maximum number of steps to decode is capped at 16. To control the computational cost and time complexity, one can also reduce the number of rollouts per beam and the beam size to $n = 2$ and $k \in [3, 4]$ respectively, as we illustrate with Llama-2 backbone.

We set generation temperatures differently for various tasks and baselines. Regarding the generation temperature $\gamma$ on Codex, for arithmetic and symbolic reasoning with PAL using deterministic beam search ($\tau = 0.0$), we find that $\gamma \in [0.4, 0.8]$ generally works well. In contrast, for commonsense reasoning with CoT, a lower temperature ($\gamma \in [0.1, 0.5]$) is more effective, likely due to the increased randomness from the free-text format. Differently, when using Llama-2 backbone, PAL generally works better with lower generation temperature $\gamma \leq 0.5$, while CoT can tolerate larger $\gamma > 0.5$ with better or comparable performance. This difference between Codex and Llama-2 may come from the different training tasks and data adopted for the two models, where the PAL reasoning is especially enhanced in Codex.

In majority voting, higher $\gamma$ is preferred to better explore the search space in reasoning, *e.g.*, $\gamma \geq 0.5$ for arithmetic reasoning. To further introduce sampling randomness in stochastic beam search for majority voting on multiple reasoning chains, we use $\alpha = 0.5$ for all datasets but different values of $\tau$ for each task. Specifically, we choose $\tau = 0.5$ for PAL and $\tau = 0.2$ for CoT, as PAL typically decomposes the reasoning problem into more steps than CoT. Here we tune $\tau$ instead of $\alpha$ to be smaller in CoT as CoT naturally contains more randomness due to its free-text formulation as we observe in practice, where a smaller $\tau$ is more efficient to balance this randomness.

In previous works, majority voting on multiple reasoning chains involves sampling $N$ (usually $\geq 20$) reasoning chains and conducting a vote to determine the final answer, which can be time-consuming. In our approach, we simply perform majority voting among the $N$ candidates in the last step of beam search from only a few times ($\leq 10$) of searching. As a result, our method does not introduce additional time complexity compared to the conventional majority voting method, although we sacrifice some diversity in the final outcomes due to the similarity of candidates within a beam.

**Prompts.** We show examples of both the generation and evaluation prompts we use on different tasks in the following tables, where we only show one instance for each case. Full prompts and detailed formulations can be found in our code.

Table 6: Examples of few-shot exemplars of generation and evaluation CoT prompts on GSM8K. We adopt 8-shot and 5-shot learning for generation and evaluation, respectively.

---

**GENERATION**

**Q**: There are 15 trees in the grove. Grove workers will plant trees in the grove today. After they are done, there will be 21 trees. How many trees did the grove workers plant today?

**A**:

There are 15 trees originally.

And there were 21 trees after some more were planted.

So 21 - 15 = 6 trees were planted.

So the answer is 6.

---

**EVALUATION**

**Q**: Allison brought some CDs online. Each CD cost $7. There was an additional charge of $4 per order for shipping costs. The total bill came to $60. How many CDs did Allison buy?

**A**:

Each CD cost 7 dollars.

# Is the above step of reasoning:

# (A) Correct

# (B) Incorrect

# The above step of reasoning is (A)

And there was an additional charge of 4 dollars.

# Is the above step of reasoning:

# (A) Correct

# (B) Incorrect

# The above step of reasoning is (A)

So the total cost of each CD is 7 + 4 = 11 dollars.

# Is the above step of reasoning:

# (A) Correct

# (B) Incorrect

# The above step of reasoning is (B), because each CD costs 7 dollars.

So 60 / 11 = 5.45.

# Is the above step of reasoning:

# (A) Correct

# (B) Incorrect

# The above step of reasoning is (B), because it cost 4 dollars for shipping costs. So the cost of CDs is 60 - 4 = 56 dollars. So Allison bought 56 / 7 = 8 CDs.

So the answer is 5.

# Is the above step of reasoning:

# (A) Correct

# (B) Incorrect

# The above step of reasoning is (A), but the value of the number of CDs is incorrect.

Table 7: Examples of few-shot exemplars of generation and evaluation PAL prompts on **GSM8K**. We adopt 9-shot and 5-shot learning for generation and evaluation, respectively.

**GENERATION**

**Q**: Olivia has $23. She bought five bagels for $3 each. How much money does she have left?

```python
def solution():
    money_initial = 23
    bagels = 5
    bagel_cost = 3
    money_spent = bagels * bagel_cost
    money_left = money_initial - money_spent
    result = money_left
    return result
```

---

**EVALUATION**

**Q**: A piece of square paper has a perimeter of 32 centimeters. Nicky's dog, Rocky, tore off 1/4 of the paper. What is the area of the remaining paper?

```python
def solution():
    perimeter = 32
    # Is the above line of code:
    # (A) Correct
    # (B) Incorrect
    # The above line of code is: (A)
    fraction_torn = 1 / 4
    # Is the above line of code:
    # (A) Correct
    # (B) Incorrect
    # The above line of code is: (A)
    area_total = (perimeter / 4) ** 2
    # Is the above line of code:
    # (A) Correct
    # (B) Incorrect
    # The above line of code is: (A), because the total area of the square
    ↪   can be calculated by (perimeter / 4) ** 2
    area_remaining = (1 - fraction_torn) * area_total
    # Is the above line of code:
    # (A) Correct
    # (B) Incorrect
    # The above line of code is: (A)
    result = area_total
    # Is the above line of code:
    # (A) Correct
    # (B) Incorrect
    # The above line of code is: (B), because the result should be
    ↪   area_remaining
    return result
    # Is the above line of code:
    # (A) Correct
    # (B) Incorrect
    # The above line of code is: (A), but the value of result is incorrect
```

Table 8: Examples of few-shot exemplars of generation and evaluation PAL prompts on **AQuA**. Here `...` represents the same evaluation script as those in the previous steps. We adopt 8-shot and 5-shot learning for generation and evaluation, respectively.

**GENERATION**

**Question**: In a flight of 600 km, an aircraft was slowed down due to bad weather. Its average speed for the trip was reduced by 200 km/hr and the time of flight increased by 30 minutes. The duration of the flight is:

**Answer Choices**: A)1 hour; B)2 hours; C)3 hours; D)4 hours; E)5 hours

```python
def solution():
    duration = Symbol('duration', positive=True)
    delay = 30 / 60
    total_disntace = 600
    original_speed = total_disntace / duration
    reduced_speed = total_disntace / (duration + delay)
    solution = solve_it(original_speed - reduced_speed - 200, duration)
    duration = solution[duration]
    result = duration
    return result
```

**EVALUATION**

**Question**: Two trains of length 150 m and 200 m are 100 m apart. They start moving towards each other on parallel tracks, at speeds 54 kmph and 72 kmph. In how much time will the trains cross each other?

**Answer Choices**: A)100/7 sec; B)80/7 sec; C)57/7 sec; D)110/7 sec; E)50/7 sec

```python
def solution():
    train_1_speed = 54 / 60
    # Is the above line of code:
    # (A) Correct
    # (B) Incorrect
    # The above line of code is: (A)
    train_2_speed = 72 / 60
    # Is the above line of code:
    # (A) Correct
    # (B) Incorrect
    # The above line of code is: (A)
    distance_between_trains = 100
    # Is the above line of code:
    # (A) Correct
    # (B) Incorrect
    # The above line of code is: (A)
    train_1_length = 150
    # Is the above line of code:
    # (A) Correct
    # (B) Incorrect
    # The above line of code is: (A)
    train_2_length = 200
    # ...
    # The above line of code is: (A)
    time_to_cross = distance_between_trains / (train_1_speed +
    ↪   train_2_speed)
    # ...
    # The above line of code is: (B), because to cross each other, the
    ↪   total distance should also contain the train length
    result = time_to_cross
    # ...
    # The above line of code is: (B), because the final result should be in
    ↪   seconds, and the value of time_to_cross is incorrect
    return result
    # ...
    # The above line of code is: (A), but the value of result is incorrect
```

Table 9: Examples of few-shot exemplars of generation and evaluation PAL prompts on **SVAMP** and **ASDiv**. Here we utilize the same prompts as they have the same task formulation. We adopt 7-shot and 5-shot learning for generation and evaluation, respectively.

**GENERATION**

**Passage**: James bought 93 red and 10 blue stickers, he used 31 red sticker on his fridge and 7 blue stickers on his laptop.

**Question**: How many red stickers does James have?

```python
def solution():
    original_red_stickers = 93
    used_red_stickers = 31
    red_stickers = original_red_stickers - used_red_stickers
    result = red_stickers
    return result
```

**EVALUATION**

**Passage**: A piece of square paper has a perimeter of 32 centimeters. Nicky's dog, Rocky, tore off 1/4 of the paper.

**Question**: What is the area of the remaining paper?

```python
def solution():
    perimeter = 32
    # Is the above line of code:
    # (A) Correct
    # (B) Incorrect
    # The above line of code is: (A)
    side_length = perimeter / 4
    # Is the above line of code:
    # (A) Correct
    # (B) Incorrect
    # The above line of code is: (A)
    area = side_length ** 2
    # Is the above line of code:
    # (A) Correct
    # (B) Incorrect
    # The above line of code is: (A)
    result = area
    # Is the above line of code:
    # (A) Correct
    # (B) Incorrect
    # The above line of code is: (B), because should calculate the
    ↪   remaining area after torn off as result
    return result
    # Is the above line of code:
    # (A) Correct
    # (B) Incorrect
    # The above line of code is: (A), but the value of result is incorrect
```

Table 10: Examples of few-shot exemplars of generation and evaluation PAL prompts on **TabMWP**. We adopt 4-shot and 5-shot learning for generation and evaluation, respectively.

## GENERATION

**Table of "Coin collections":**

Name | Number of coins

Braden | 76 \\ Camilla | 94 \\ Rick | 86

Mary | 84 \\ Hector | 80 \\ Devin | 83

Emily | 82 \\ Avery | 87

**Question**: Some friends discussed the sizes of their coin collections. What is the mean of the numbers?

```python
def solution():
    number_of_coins_for_different_person = [76, 94, 86, 84, 80, 83, 82, 87]
    mean_of_the_numbers = sum(number_of_coins_for_different_person) /
    ↪   len(number_of_coins_for_different_person)
    result = mean_of_the_numbers
    return result
```

## EVALUATION

**Table of "Roller coasters per amusement park":**

Stem | Leaf

1 | 0, 0, 1, 6, 8, 9 \\ 2 | 4, 4, 5, 7, 8, 8

3 | 1, 2, 4, 4, 9, 9 \\ 4 | 2, 3, 5, 6, 8, 9, 9

**Question**: Rodrigo found a list of the number of roller coasters at each amusement park in the state. How many amusement parks have fewer than 40 roller coasters?

```python
def solution():

    number_of_roller_coasters_per_amusement_park = [10, 14, 14, 15, 16, 18, 19, 20, 24, 25, 26, 28, 29, 29, 29,
    ↪   30, 34, 35, 36, 39, 40, 40, 40, 41, 42, 43, 44, 44, 45, 45, 46, 46, 47, 48, 48, 49, 49, 49, 50, 50, 51,
    ↪   51, 52, 52, 53, 53, 54, 54, 55, 55, 56, 57, 57, 58, 58, 59, 59, 60, 60, 61, 61, 62, 62, 63, 63, 64,
    ↪   64, 65, 65, 66, 66, 67, 67, 68, 68, 69, 69, 70, 70, 71, 71, 72, 72, 73, 73, 74, 74, 75, 75, 76, 77,
    ↪   77, 78, 78, 79, 79, 80, 80, 81, 81, 82, 82, 83, 83, 84, 84, 85, 85, 86, 86, 87, 87, 88, 88, 89, 89, 90,
    ↪   90, 91, 91, 92, 92, 93, 93, 94, 94, 95, 95, 96, 96, 97, 97, 98, 98, 99, 99]
    # Is the above line of code:
    # (A) Correct
    # (B) Incorrect
    # The above line of code is: (B), beacuse values in the rows of Stem and Leaf represent the decimal and
    ↪   individual digits, respectively

    number_of_amusement_parks_with_fewer_than_40_roller_coasters = 0
    # Is the above line of code:
    # (A) Correct
    # (B) Incorrect
    # The above line of code is: (A), because this is to initialize the
    ↪   number_of_amusement_parks_with_fewer_than_40_roller_coasters
    for number_of_roller_coasters in
    ↪   number_of_roller_coasters_per_amusement_park:
        if number_of_roller_coasters < 40:
            number_of_amusement_parks_with_fewer_than_40_roller_coasters +=
            ↪   1
            # Is the above line of code:
            # (A) Correct
            # (B) Incorrect
            # The above line of code is: (A), but the value of
            ↪   number_of_roller_coasters_per_amusement_park is incorrect
    result = number_of_amusement_parks_with_fewer_than_40_roller_coasters
    # Is the above line of code:
    # (A) Correct
    # (B) Incorrect
    # The above line of code is: (A), but the value of
    ↪   number_of_amusement_parks_with_fewer_than_40_roller_coasters is
    ↪   incorrect
    return result
    # ...
    # The above line of code is: (A), but the value of result is incorrect
```

Table 11: Examples of few-shot exemplars of generation and evaluation PAL prompts on `Date Understanding` from **Big-Bench**. We adopt 6-shot and 3-shot learning for generation and evaluation, respectively.

---

**GENERATION**

**Q**: 2015 is coming in 36 hours. What is the date one week from today in MM/DD/YYYY?

```python
def solution():
    # If 2015 is coming in 36 hours, then today is 36 hours before.
    today = datetime(2015, 1, 1) - relativedelta(hours=36)
    # One week from today,
    one_week_from_today = today + relativedelta(weeks=1)
    # The answer formatted with %m/%d/%Y is
    result = one_week_from_today.strftime('%m/%d/%Y')
    return result
```

---

**EVALUATION**

**Q**: Yesterday was April 30, 2021. What is the date yesterday in MM/DD/YYYY?

```python
def solution():
    # Yesterday was April 30, 2021.
    yesterday = datetime(2021, 4, 30)
    # Is the above line of code:
    # (A) Correct
    # (B) Incorrect
    # The above line of code is: (A)
    # Yesterday,
    yesterday = yesterday - relativedelta(days=1)
    # Is the above line of code:
    # (A) Correct
    # (B) Incorrect
    # The above line of code is: (B), because the date of yesterday was
    ↪  already got and this is step is redundant
    # The answer formatted with %m/%d/%Y is
    result = yesterday.strftime('%m/%d/%Y')
    # Is the above line of code:
    # (A) Correct
    # (B) Incorrect
    # The above line of code is: (A), but the value of yesterday is
    ↪  incorrect
    return result
    # Is the above line of code:
    # (A) Correct
    # (B) Incorrect
    # The above line of code is: (A), but the value of result if incorrect
```

Table 12: Examples of few-shot exemplars of generation and evaluation PAL prompts on `Object Counting` from **Big-Bench**. We adopt 4-shot and 1-shot for generation and evaluation, respectively.

---

**GENERATION**

**Q**: I have a drum, a flute, a clarinet, a violin, four accordions, a piano, a trombone, and a trumpet. How many musical instruments do I have?

```python
def solution():
    musical_instruments_to_count = {
        'drum': 1,
        'flute': 1,
        'clarinet': 1,
        'violin': 1,
        'accordion': 4,
        'piano': 1,
        'trombone': 1,
        'trumpet': 1
    }
    num_musical_instruments = sum(musical_instruments_to_count.values())
    result = num_instruments
    return result
```

---

**EVALUATION**

**Q**: I have a peach, an onion, two garlics, a nectarine, a yam, a carrot, a potato, a head of broccoli, a cabbage, a banana, and a stalk of celery. How many vegetables do I have?

```python
def solution():
    # note: I'm not counting the peach, nectarine, banana, or celery as
    ↪   vegetables
    vegetables_to_count = {
        'onion': 1,
        # ...
        # The above line of code is: (A)
        'garlic': 2,
        # ...
        # The above line of code is: (A)
        'yam': 1,
        # ...
        # The above line of code is: (A)
        'carrot': 1,
        # ...
        # The above line of code is: (A)
        'potato': 1,
        # ...
        # The above line of code is: (A)
        'broccoli': 1,
        # ...
        # The above line of code is: (A)
        'cabbage': 1
        # ...
        # The above line of code is: (A)
    }
    # ...
    # The above line of code is: (B), because celery is vegetable
    num_vegetables = sum(vegetables_to_count.values())
    # ...
    # ... (A), but the value of vegetables_to_count is incorrect
    result = num_vegetables
    # ...
    # ... (A), but the value of num_vegetables is incorrect
    return result
    # ...
    # ... (A), but the value of result is incorrect
```

Table 13: Examples of few-shot exemplars of generation and evaluation CoT prompts on **CSQA**. We adopt 7-shot and 3-shot learning for generation and evaluation, respectively.

---

**GENERATION**

**Q**: What do people use to absorb extra ink from a fountain pen?

**Answer Choices**: (a) shirt pocket; (b) calligrapher's hand; (c) inkwell; (d) desk drawer; (e) blotter

**A**:

The answer must be an item that can absorb ink.

Of the above choices, only blotters are used to absorb ink.

So the answer is (e).

---

**EVALUATION**

**Q**: John's took his appointment book with him to a checkup but couldn't find it anywhere. He looked and he looked but he didn't know where it was. It wasn't until he bought a new one, went home, took a shower that he found it. Where was it?

**Answer Choices**: (a) office supply store; (b) wallet; (c) doctor's office; (d) handbag; (e) pocket

**A**:

The answer must be a place where John could have put his appointment book.

# Is the above step of reasoning:

# (A) Correct

# (B) Incorrect

# The above step of reasoning is: (B), because it fails to reason out that John will check the place when taking a shower (taking off clothes)

Of the above choices, only wallet makes sense.

# Is the above step of reasoning:

# (A) Correct

# (B) Incorrect

# The above step of reasoning is: (B), because wallet is too small to put his appointment book, and it cannot be the only one which makes sense (pocket is also plausible)

So the answer is (b).

# Is the above step of reasoning:

# (A) Correct

# (B) Incorrect

# The above step of reasoning is: (A), but the previous step is incorrect

---

Table 14: Examples of few-shot exemplars of generation and evaluation CoT prompts on **StrategyQA**. We adopt 6-shot and 4-shot learning for generation and evaluation, respectively.

---

**GENERATION**

**Q**: Do hamsters provide food for any animals?

**A**:

Hamsters are prey animals.

Prey animals are food for predators.

Thus, hamsters provide food for some animals.

So the answer is yes.

---

**EVALUATION**

**Q**: While viewing "Scary Movie" is the viewer likely to experience an increase in adrenaline?

**A**:

Adrenaline is a hormone that is released when the body is under stress.

# Is the above step of reasoning:

# (A) Correct

# (B) Incorrect

# The above step of reasoning is: (A)

Scary movies are designed to make the viewer feel stressed.

# Is the above step of reasoning:

# (A) Correct

# (B) Incorrect

# The above step of reasoning is: (B), because Scary Movie is a film that is a comedy take on horror, intended to make viewers laugh

Thus, the viewer is likely to experience an increase in adrenaline.

# Is the above step of reasoning:

# (A) Correct

# (B) Incorrect

# The above step of reasoning is: (A), but the previous step is incorrect

So the answer is yes.

# Is the above step of reasoning:

# (A) Correct

# (B) Incorrect

# The above step of reasoning is: (A), but the previous step is incorrect

---

Table 15: Examples of few-shot exemplars of generation and evaluation CoT prompts on `Sports Understanding` on **Big-Bench**. We adopt 8-shot and 2-shot learning for generation and evaluation, respectively.

---

**GENERATION**
**Q**: Is the following sentence plausible?
"KYLE PALMIERI WAS CALLED FOR SLASHING."
**A**:
Kyle Palmieri is a hockey player.
Being called for slashing is part of hockey.
So the answer is yes.

---

**EVALUATION**
**Q**: Is the following sentence plausible?
"ROBIN LEHNER LAUNCHED THE DESPERATION HEAVE"
**A**:
Robin Lehner is a hockey player.
# Is the above step of reasoning:
# (A) Correct
# (B) Incorrect
# The above step of reasoning is: (B), because Robin Lehner is a goaltender which is different from a player
Launching the desperation heave is part of hockey.
# Is the above step of reasoning:
# (A) Correct
# (B) Incorrect
# The above step of reasoning is: (B), because launching the desperation heave is for player, not goaltender
So the answer is yes.
# Is the above step of reasoning:
# (A) Correct
# (B) Incorrect
# The above step of reasoning is: (A), but the previous step is incorrect

---

