# OpenReview forum: "Self-Evaluation Guided Beam Search for Reasoning"
_NeurIPS.cc/2023/Conference — NeurIPS 2023 poster_

### Official Review · Reviewer_wb7Q · 2023-07-02

**Soundness:** 4 excellent
**Presentation:** 2 fair
**Contribution:** 3 good
**Rating:** 6
**Confidence:** 2

**Summary:**

The paper proposes using a fine-grained self-evaluation method to refine multi-step reasoning inference of the Large Language Models. The authors propose a novel promting approach that integrates self-evaluation guidance through stochastic beam search. Concretely, to guide the beam search process, they employ LLM self-evaluation as an automatic criterion to offer the guiding signal. And they incorporate temperature-controlled randomness to optimize the traditional beam search algorithm in order to balance the quality-diversity trade-off for better reasoning chains. Experimental results of their methods on various arithmetic, symbolic, and commonsense reasoning tasks show considerable improvements.

**Strengths:**

1. The authors propose a novel and general multi-step decoding method (a beam search variant) which employs LLM self-evaluation and temperature-controlled randomness. The method aims to improve LLM performance on reasoning tasks.
2. Extensive experiments are conducted on a wide range of reasoning benchmarks and considerable improvements are achieved. Besides, comprehensive ablations and running examples are presented to further demonstrate the effectiveness of their approach.

**Weaknesses:**

The paper should be more self-contained. For example, the notion of self-evaluation is first used in the introduction section without a concise explanation, which makes it a bit hard to follow. And the details of the evaluation benchmark are not introduced (arithmetic, symbolic, and commonsense reasoning tasks).

**Questions:**

Authors should concisely introduce the notions when they first use it (e.g., self-evaluation) to make the paper more self-contained. Besides, the details of the different evaluation benchmarks and compared baseline methods should be concisely introduced in the experiment section as well.

**Limitations:**

The authors have adequately addressed the limitations of their work.

---

> ### Author Rebuttal · Authors · 2023-08-10
>
> We appreciate your constructive feedback.!
>
> > The paper should be more self-contained.
>
> We recognize the importance of clearly defining terms and notions when initially introduced.
>
> For "self-evaluation", the notion technically means using the LLM to evaluate its own generation.
> This includes prompting it to do direct correctness assessment and extracting the output likelihood as the implicit generation confidence.
>
> In the revised manuscript, we will ensure these terms are succinctly clarified upon the first mention.
>
>
>
> > The details of the different evaluation benchmarks and compared baseline methods should be concisely introduced in the experiment section as well.
>
> Thanks for the suggestions. We briefly introduce the benchmarks and baseline methods as follows. We will provide these details in the final paper.
>
> - Evaluation benchmarks
>     * **GSM8K**: math word problems (test - $1,319$)
>     * **AQuA**: algebraic word problems with multiple choice (test - $253$)
>     * **SVAMP**: math word problems with varying structures (test - $1,000$)
>     * **ASDiv**: diverse math word problems (test - $2,096$)
>     * **TabMWP**: table-based math word problems (test - $7,686$)
>     * **Date Understanding**: inferring a date from a given context (test - $369$)
>     * **Object Counting**: answering questions about the number of objects belonging to a certain type (test - $1,000$)
>     * **CommonsenseQA**: commonsense questions about the world involving complex semantics that often require prior knowledge (train/dev/test - $9,741$/$1,221$/$1,140$)
>     * **StrategyQA**: to infer a multi-hop strategy to answer questions (test - $2,290$)
>     * **Sports Understanding**: determining whether a sentence relating to sports is plausible or implausible (test - $1,000$)
>
> - Baseline methods
>     * **PAL**: program-aided language model – using Python program as intermediate reasoning
>     * **CoT**: chain-of-thought – using free-text “rationale” as intermediate reasoning

---

> > ### Comment · Reviewer_wb7Q · 2023-08-20
> >
> > Thanks for the response. I would like to keep my rating.

---

### Official Review · Reviewer_WznY · 2023-07-03

**Soundness:** 1 poor
**Presentation:** 3 good
**Contribution:** 1 poor
**Rating:** 3
**Confidence:** 5

**Summary:**

The paper proposes a self-evaluation enhanced beam search for using LLMs on multi-step reasoning tasks. Specifically, in addition to standard beam search that only scores a beam based on the probability, the method further adjusts the score with a self-evaluation-based faithfulness score reflecting the correctness of intermediate reasoning steps. The score is obtained by LLMs using few-shot prompting and is calculated for each rollout. The paper applies this approach on several arithmetic reasoning tasks, symbolic reasoning tasks, and common sense reasoning tasks. The results suggest that the proposed beam search variant achieves better performance compared to greedy decoding. The paper also includes a series analysis on the effectiveness of calibration, sensitivity to hyper-parameters, etc.

**Strengths:**

The proposed approach shows an application of self-calibration for multi-step reasoning tasks.

The experiments cover three tasks and multiple datasets. The paper also includes a series of analyses providing more details on the effectiveness of calibration for reasoning tasks.

The paper is well-written and easy to follow.

**Weaknesses:**

1. The most critical issue is that the comparison in this paper is misleading.

The paper draws the majority of attention to the comparison between the proposed beam-search-based approach against greedy decoding. It’s understood that both the greedy decoding and the beam search produce one single chain. But the proposed approach uses *beam search* and self-evaluation (which is also done by few-shot prompting). This is way more expensive than greedy decoding.

Specifically,  based on the hyper-parameter in Appendix A.3, the model uses beam size (k=5, and use n=16 rollouts for each beam). Assuming we evaluate a problem with steps T=3. In a rough estimation, this approach needs to call the API to generate 5 * 16 * (3 - 1) * 2 = 320 completions (there is a factor 2 because the approach also uses LLM to self-evaluate each rollout, and based on the appendix, the length of the evaluation prompt is at the same magnitude as the length of the few-shot prompt). Not that this estimation follows the way that OpenAI charges API calls, which charges both prompt and completion tokens, see https://openai.com/pricing. In contrast, greedy decoding only requires running 1 completion. There’s a distinct gap in the computation cost (320 vs 1 measured by API cost). Similarly, the computation cost in the multiple reasoning chains setup is also different.

In short, the experiments and the interpretation of results need substantial refactoring. At least the paper should compare the proposed approach against baselines that uses the same amount of computation instead of producing the same number of reasoning chains. As a result, the improvements in Table 1 are inflated (E.g., considering PAL with 8 samples for self-consistency decoding can get ~78% performance on GSM8K as shown in Figure 12 in PAL).


2. At the same time, the core technical contribution of the proposed approach is a straightforward extension of beam search based on self-evaluation, it is hard to judge the empirical merits of the proposed approach without a fair, clear, and thorough comparison.


**Questions:**

Can authors provide the exact cost (measured by API cost, or tokens being scored) for PAL and the proposed approach in the single chain setting of Table 1?


When calculating $P_{LM}$, is the probability of a reasoning step normalized by the number of tokens?



**Limitations:**

Some limitations are discussed at the end of the paper; no obvious negative societal impact as far as I know.

---

> ### Author Rebuttal · Authors · 2023-08-10
>
> Thank you for the insightful feedback.
>
> > Can authors provide the exact cost (measured by API cost, or tokens being scored) for PAL and the proposed approach in the single chain setting of Table 1?
>
> Sure. We provide a detailed comparison regarding the cost in Table 1 of the pdf.
> Please refer to the global response for the discussion on this.
>
>
> > The most critical issue is that the comparison in this paper is misleading.
>
> We acknowledge the importance of fair comparison in terms of computation costs.
> We provide further discussion on this in the global response.
>
> However, in our original paper, we follow the previous works in reasoning to compare methods using the same number of reasoning chains for self-consistency.
>
> Furthermore, our method also enhances the effect of self-consistency by increasing the performance upper bound when scaling the $N$ in majority voting.
> Theoretically, the upper bound of majority voting (Tangian. 2020) is determined by the number of questions with $p > 0.5$, where $p$ represents the probability of correct prediction for each question (Condorcet's jury theorem).
> Increasing the computational cost by increasing the members of the majority voting cannot improve this upper bound.
> On the other hand, our proposed method modifies this $p$ to increase the number of questions with $p > 0.5$, thus increasing the upper bound.
> By increasing $p$, the effect of a small finite $N$ in the majority voting also improves with our method, since the convergence rate of the majority voting increases as $p$ increases.
> In the global response, Figure 1 shows the empirical estimation of how our method improves this $p$ significantly.
> This demonstrates a bonus of our method when combined with majority voting.
> We will conduct comprehensive experiments on open-source models to further analyze the upper bounds of our method.
>
>
> > At least the paper should compare the proposed approach against baselines that uses the same amount of computation instead of producing the same number of reasoning chains.
>
> We appreciate your valuable advice.
> As discussed in the global response for Table 1, our generation cost is equivalent compared to the self-consistenc.
> Also, our method shows promising enhancement when combined with majority voting.
> We will conduct comprehensive experiments on this using the open-source LMs.
>
>
> > When calculating, is the probability of a reasoning step normalized by the number of tokens?
>
> Yes, the step probability is the geometric mean of the token probabilities.
> This is to mitigate the difference in probabilities caused by the step length.
>
>
>
> ---
> Analytical Theory of Democracy (Andranik Tangian. 2022)
>
> Condorcet's jury theorem (Wikipedia)

---

> > ### Comment · Reviewer_WznY · 2023-08-11
> >
> > I appreciate the authors' clarification.
> >
> > IIUC, as stated in the attachment pdf. For the single-chain results, the average cost estimated by number of output tokens only for PAL+greedy (B1) decoding is 600, and for the proposed approach (O1) is 16\*60+(5-1)\*5\*16\*60 = 20160, which is 33X cost measured by output tokens. With the same computation, PAL is actually on par with the proposed approach. Furthermore, the paper is using API, which actually does not cache prompt tokens, and the proposed approach is even much more computationally intensive (this roughly matches the estimation mentioned in my original response above).
> >
> > >  However, in our original paper, we follow the previous works in reasoning to compare methods using the same number of reasoning chains for self-consistency.
> >
> > I am not sure if I can be convinced by this. A direct comparison between two approaches requiring computation budget at two significantly different magnitudes still seems a bit off to me, personally. The past work also exhibits such an issue (in a little bit less severe way), but the community is being more aware of this issue as well [1].
> >
> > Overall, I agree that using this technique could increase the upper bound when having enough computation, but I feel the whole experiment section and the interpretation of results need to be substantially refactored. It is hard for me to form a judgment without seeing a substantially revised version of the paper.
> >
> > [1] Demystifying GPT Self-Repair for Code Generation. Olausson et al., 2023.

---

> > > ### Author Response · Authors · 2023-08-14
> > >
> > > We acknowledge your points regarding the computation costs. We provided a more thorough analysis in our [comment](https://openreview.net/forum?id=Bw82hwg5Q3&noteId=b85ueeEcFP) under the global response.

---

> > > > ### Comment · Reviewer_WznY · 2023-08-17
> > > >
> > > > Thanks for the response and the results update, especially for the new results with LLAMA2 under fair comparison.
> > > >
> > > > There are lots of new details and new results that are added in the rebuttal phase and discussion phase. There seems to be substantial effort to properly incorporate them in the paper, (e.g., the LLAMA2 results are only on the GSM8K dataset, the setting needs to be propagated to other datasets as well as the analysis section).
> > > >
> > > > It might just be me, but it is hard for me to form a judgment without seeing major revision. Therefore, I am inclined to keep my score unchanged, but I won't object to acceptance if other reviewers are strongly in favor of it.

---

### Official Review · Reviewer_TWEq · 2023-07-03

**Soundness:** 3 good
**Presentation:** 2 fair
**Contribution:** 3 good
**Rating:** 6
**Confidence:** 4

**Summary:**

The authors propose using beam search to guide each reasoning step in chain-of-thought prompting. Functionally, the paper is quite similar to the concurrent "Tree of Thoughts" work from Yao et al. (2023).

**Strengths:**

Overall, it's a generally solid work. The experiments are reasonable and fairly comprehensive; the method, while not groundbreaking, is the natural next step of a lot of work in this area and is well-motivated; its figures are good and convey its message well; the results are generally positive - it would be very surprising to me if they weren't.

**Weaknesses:**

Primary concerns:
- Because the self-evaluation requires extra inference, it's not entirely obvious that all of the comparisons are fair, so it would be useful to see some discussion around this. There's also the overhead from being unable to cache preceding tokens, but that's an artifact of using an API. The paper doesn't seem to discuss details like the number of tokens generated or the number of beams used, which in general, make comparison hard.
- The title is very confusing - this paper has little to do with "decomposition" since every step taken is sequential. This should really be fixed as it implies a very different message than the one actually made by the paper. Normally this wouldn't be a substantial issue, but in this case, it is especially confusing.
- The writing could really use work - the phrasing is awkward and often nonstandard, like the first sentence of the abstract "We endow Large Language Models (LLMs) with fine-grained self-evaluation to refine multi-step reasoning inference." In general, the paper reads fairly awkwardly and is unnecessarily wordy.

Other concerns:
- The novelty of the paper seems to be in the use of model-probability-guided beam search itself, but in my view, the paper somewhat oversells the novelty of using model self-evaluation for selection. Even within the code domain there's lots of work on this, such as "Coder Reviewer Reranking for Code Generation" by Zhang et al. (2022). Plus, there is work on evaluating uncertainty with prompting which would probably even be better suited for this, like the concurrent "Semantic Uncertainty: Linguistic Invariances for Uncertainty Estimation in Natural Language Generation" by Kuhn et al. (2023). This is not to say that the paper is not novel, but the main novelty seems to be the use of self-guided beam search in general and not the "Self-Evaluation as Faithfulness Constraint."
- This method isn't really possible with recent closed language models that don't expose logprobs (as mentioned), and it's not obvious how to integrate this. I also don't really feel that it's fair to refer to this as faithfulness.

**Questions:**

How do you ensure the comparison with the other models is fair? How many beams do you use? For k beams, is it more useful to sample 2*k chains of thought?

**Limitations:**

Reasonable discussion.

---

> ### Author Rebuttal · Authors · 2023-08-10
>
> We thank the reviewer for their insightful comments.
>
> > Because the self-evaluation requires extra inference, it's not entirely obvious that all of the comparisons are fair, so it would be useful to see some discussion around this.
>
> Thanks for the suggestion. Please see the global response for this.
>
>
> > How many beams do you use? For k beams, is it more useful to sample 2*k chains of thought?
>
> Kindly refer to the implementation details in Appendix A.3.
> We also provide additional discussion on this in the global response.
>
>
> > The title is very confusing - this paper has little to do with "decomposition" since every step taken is sequential.
>
> We appreciate your concern regarding the term "decomposition".
>
> We adopt “decomposition” to highlight that our method calibrates the reasoning chain in a “step-by-step” manner, as illustrated in Figure 1 and discussed in the first part of Section 4 (Related Work).
> We admit that this may introduce ambiguity, especially since the reasoning process leans more towards sequential generation rather than a hierarchical decomposition of the problem.
> We will revisit the title to better align it with the paper's content.
>
>
> > This method isn't really possible with recent closed language models that don't expose logprobs (as mentioned), and it's not obvious how to integrate this.
>
> We illustrated how to extract scores from closed-source LMs, such as GPT-4, in Appendix A.2 under GPT-Experiments.
> While the exact logits -- to get the step log-likelihood -- are not available, our case study suggests that GPT-4 scoring serves as a promising metric for evaluating reasoning quality.
> Practically, GPT-4 scores can be directly utilized as evaluation outcomes when implementing our method.
> We can also leverage the textual feedback from GPT-4 to refine these scores, catering to diverse reasoning criteria.
>
>
> > This is not to say that the paper is not novel, but the main novelty seems to be the use of self-guided beam search in general and not the "Self-Evaluation as Faithfulness Constraint."
>
> Thanks for raising your concerns about the novelty and the word “faithfulness”.
>
> We acknowledge the contributions of previous research, including the conditional likelihood-based consensus solicitation of Zhang et al (2022) and the uncertainty estimation method introduced by Kuhn et al (2023).
> We will cite these works and compare them with the LLM self-evaluation.
>
> To clarify, our claim isn't about being the first to introduce self-evaluation.
> Our primary emphasis is on constructing the self-guided beam search framework.
>
> Regarding "faithfulness", our original goal is to highlight the calibration ability of self-evaluation based on the consensus between the assessment and the generation.
> An alternative way to feature this can be "consistency" or "fidelity".
>
> We will consider the term used and refine the paper to make our contribution clearer.
>
>
> > The writing could really use work
>
> Thanks for the constructive advice.
> We will improve the clarity in the final version.

---

> > ### Comment · Reviewer_TWEq · 2023-08-10
> >
> > A few clarifications to follow up:
> > 1. Is N (as used in the optional PDF rebuttal) also 40 for your approach, as implied by "multiple reasoning chains (N = 40)"? If so, you are generating 40 / 5 * (16 * 60 + (5 - 1) * 5 * 16 * 60), or approximately 161,000 tokens per problem, correct? That would roughly be an order of magnitude more than PAL + SC at 24,000. To reiterate my question on a fair comparison concretely: given the same number of generated tokens, does this method outperform PAL + SC?
> > 2. Decomposition has a fairly standard meaning when discussing reasoning, and it does not mean doing things in a step-by-step way. When you say you'll revisit the title, can you please clarify whether you intend to correct this title, and if so, how.
> > 3. I don't want to get into a prolonged discussion about faithfulness, because the above two questions are much more important to me, but given how "faithfulness" is typically used in reasoning papers, it doesn't seem particularly relevant here.

---

> > > ### Author Response · Authors · 2023-08-14
> > >
> > > > given the same number of generated tokens, does this method outperform PAL + SC?
> > >
> > > We appreciate your suggestions on the cost analysis.
> > > In our latest results in the [comment](https://openreview.net/forum?id=Bw82hwg5Q3&noteId=b85ueeEcFP) under the global response, we have shown that our method outperforms PAL + SC given the comparable number of tokens.
> > > Please refer to it for details.
> > >
> > > > can you please clarify whether you intend to correct this title, and if so, how
> > >
> > > Thanks for the clarifying question. We will change the title to "Self-Evaluation Guided Beam Search for Reasoning" to reflect our contributions more clearly.
> > > We will also update the paper contents related to the new title accordingly.
> > >
> > > > faithfulness
> > >
> > > We appreciate your concern about the appropriate use of "faithfulness". We will change it to "self-evaluation score" to avoid confusion.

---

> > > > ### Comment · Reviewer_TWEq · 2023-08-14
> > > >
> > > > Thank you for the new experiments. I find these results fairly convincing of the claim that there are configurations where this approach outperforms PAL+SC, at least for smaller LMs, which was my primary concern. I also think the revised title will be much clearer. I have accordingly raised my score.

---

### Official Review · Reviewer_fpKX · 2023-07-04

**Soundness:** 3 good
**Presentation:** 2 fair
**Contribution:** 3 good
**Rating:** 6
**Confidence:** 4

**Summary:**

This paper proposes to improve the reasoning capability of large language models (LLMs) by decomposing the reasoning task into multiple steps and then performing self-evaluation guided beam search to find the optimal reasoning chain. On a set of arithmetic reasoning, symbolic reasoning, and commonsense reasoning tasks, the proposed method outperforms baselines in most cases. The paper also provided analysis regarding the quality-diversity trade-off in the generation of the reasoning chain.

**Strengths:**

1. This paper proposed a novel and effective stochastic beam search which searches for multi-step reasoning. The approach integrates the LLM probability distribution and the self-evaluation faithfulness score.
2. The proposed method is evaluated on multiple datasets. The discussion and analyses are helpful.

**Weaknesses:**

1. The proposed method is very costly, as it requires prompting the LLM for multiple times (when searching for next steps and for self evaluation). Its effectiveness also depends on the LLM search space, e.g., when an LLM is very confident with their (incorrect) reasoning, the potential of locating the correct steps in beam search will become small.

2. Some details are missing or require clarification:
- How did the framework define one "step" in each reasoning dataset? For example, is every step a single statement for PAL and a complete sentence (i.e., ended with a period) for CoT?
- Figure 4(a) is not referred to in the paper. It is unclear based on which dataset the two plots were drawn. The numbers on the plots also need more explanation. For example, in the left figure, why is PAL+SC so stable when changing the LM temperature (which I believe would end up with very different sets of samples in SC)? Similarly in the right figure, why is CoT+SC changing if it does not involve beam search at all (and hence should be agnostic to the change of the sampling temperature in Eq 4)?

3. Baseline: how does the proposed method compare with self-refinement (Madaan et al., 2023), which, instead of searching for better steps during reasoning, iteratively self-refines the reasoning output?

**Questions:**

Please address my concerns and questions in Weaknesses.

**Limitations:**

The paper included a limitations section.

---

> ### Author Rebuttal · Authors · 2023-08-10
>
> We appreciate the reviewer's insightful suggestions!
>
>
> > The proposed method is very costly
>
> We recognize the substantial cost associated with beam search.
> Please refer to the global response for the discussion on this.
>
>
> > Its effectiveness also depends on the LLM search space, e.g., when an LLM is very confident with their (incorrect) reasoning, the potential of locating the correct steps in beam search will become small.
>
> We appreciate your observation regarding the constraints within the LLM search space.
> However, we posit that this limitation is intrinsic to few-shot prompting techniques.
> Our method incorporates self-evaluation to guide and calibrate the generation process.
> This is based on the insight that LLMs are better at assessing the correctness of an answer than directly producing it (Kadavath et al. 2022).
>
>
> > How did the framework define one "step" in each reasoning dataset?
>
> This depends on the reasoning format.
> For PAL, reasoning takes the form of a Python program, with each complete statement symbolizing a "step". We employed a regex-based method to segment the program into these steps.
> For CoT, the reasoning is in unstructured text. We directly tokenize the reasoning chain into sentences, with each representing a "step".
>
> For further implementation nuances, kindly check the code in supplementary materials.
>
>
> > Figure 4(a) is not referred to in the paper.
>
> Thanks for highlighting the concern.
>
> Both plots were constructed based on GSM8K.
> The lines “PAL+SC” and “CoT+SC” represent results reported in the PAL paper.
> We display them as reference baselines.
> In this case, “PAL+SC” and “CoT+SC” are illustrated with fixed LM temperature settings.
> So yes, they should be stable and won't change in the plots.
>
> We will clarify this in the caption in our revised version.
>
>
> > How does the proposed method compare with self-refinement (Madaan et al., 2023), which, instead of searching for better steps during reasoning, iteratively self-refines the reasoning output?
>
> Thanks for bringing up the work on self-refinement.
>
> This approach conducts instance-level refinement, while our method works on the granular aspects.
> Therefore, our advantages differ based on the problem type and the scale of the output.
> For example, while self-refinement offers a holistic perspective to diagnose and correct flaws, it may struggle to detect minor yet critical mistakes in longer generations.
>
> Furthermore, it's worth noting that in the rapidly evolving field of AI research, it is indeed challenging to keep abreast of the very latest methods.
> We acknowledge the importance and relevance of the self-refinement method, and since a direct comparison might be out of scope for this paper, we will consider discussing it regarding the future direction if space permits.
>
>
> ---
> Language Models (Mostly) Know What They Know (Kadavath et al. 2022)

---

> > ### Comment · Reviewer_fpKX · 2023-08-13
> >
> > Thanks for your clarification! I hope in the next version you can make the main-text writing more self-contained.
> >
> > Fair comparison -- Clearly this is the major concern of all reviewers. To make things easier, can you make a new Table 1 to replace the current one in your paper draft, which includes not only the model performance and each model's cost? I will change my score accordingly.
> >
> > My additional comments:
> > 1. Table 1 in your attached PDF: Thanks for helping us understand the cost calculation. However,
> > - I want to make sure that the numbers in your caption are consistent with your experimental setup in the main-text Table 1. For example, are the SC results all based on N=40? It is also unclear how you set `x_B=600` and `x_O=60` (when `T=5`, is this saying that the reasoning chain generated by your approach roughly has a length of `x_O * T = 300` which is half lengthy as PAL?)
> > - The estimation in your caption only talks about the generation cost, but what about the prompt cost? For B2, the SC case should have #input tokens being C, not NC, because running the SC experiment will only require setting up the number of completions under one run of prompting.
> > - "caching past hidden states during inference if we use open-sourced models" -- This I agree, but for OpenAI APIs this is not feasible yet, so the cost needs to be clarified separately.
> >
> > 2. I have forgotten to ask -- why are most results of PAL-SC missing in your main-text Table 1? I hope they can be filled for a more fair comparison.

---

> > > ### Author Response · Authors · 2023-08-14
> > >
> > > > To make things easier, can you make a new Table 1 to replace the current one in your paper draft, which includes not only the model performance and each model's cost?
> > >
> > > Thanks for the suggestion.
> > > We have provided experiments on open-source models in the [comment](https://openreview.net/forum?id=Bw82hwg5Q3&noteId=b85ueeEcFP) under the global response.
> > > We will use this format to reorganize the results in Table 1 in our paper.
> > >
> > > We have also computed the actual number of tokens (including input and output) using Codex.
> > > For reference, below are the API cost (\# tokens) per instance on GSM8K.
> > > The cost of PAL+SC is not exactly $NC$ as we sent batch requests for API calls.
> > > Instead, the cost is divided by the batch size in practice.
> > >
> > > | Methods | \# Input Tokens | \# Output Tokens |   Accuracy |
> > > |:-:|:-:|:-:|:-:|
> > > | single-chain |
> > > |  PAL    |      1.6k       |      409.1       |  72.0 |
> > > |  Ours   |      1.5m       |       16k        |  80.2 |
> > > | multiple-chain (N=40) |
> > > |  PAL+SC |      13k        |       18k        |  80.4 |
> > > |  Ours   |      12m        |      640k        |  85.5 |
> > >
> > > > I want to make sure that the numbers in your caption are consistent with your experimental setup in the main-text Table 1. For example, are the SC results all based on N=40? It is also unclear how you set x_B=600 and x_O=60 (when T=5, is this saying that the reasoning chain generated by your approach roughly has a length of x_O * T = 300 which is half lengthy as PAL?)
> > >
> > > Thanks for the clarifying question.
> > > For GSM8K, we always use $N=40$ for baselines and our method.
> > > And we choose $x_B=600$ as the maximum length following the implementation code of PAL.
> > > The max $T$ was set as $10$ in our method, we used $T=5$ in our previous computation as an average number of steps.
> > > In our latest results in the [comment](https://openreview.net/forum?id=Bw82hwg5Q3&noteId=b85ueeEcFP) under the global response, we are computing the actual tokens thus it should be accurate.
> > >
> > > > The estimation in your caption only talks about the generation cost, but what about the prompt cost?
> > >
> > > Thanks for highlighting the prompting cost.
> > > Specifically, we have input length of prompts $C \approx 1.5$k.
> > > Given the average number of steps $T=5$, the beam size $k=5$, and the number of rollouts $n=16$,
> > > the average input cost of our method is $2nC+2(T-1)knC=672\times 1.5$k $\approx 1$m tokens.
> > > For PAL+SC, as we use mini-batch requests with a batch size of $5$, the cost can be calculated as $NC/5=8\times 1.5$k$=12$k.
> > > In our latest results using open-source models, we have substantially reduced the prompt cost of our method with smaller $n$ and by caching past hidden states.
> > > Please kindly refer to the [comment](https://openreview.net/forum?id=Bw82hwg5Q3&noteId=b85ueeEcFP) under the global response for details.
> > >
> > > > For B2, the SC case should have #input tokens being C, not NC, because running the SC experiment will only require setting up the number of completions under one run of prompting.
> > >
> > > We appreciate your concern about the cost of self-consistency.
> > > It is worth noting that in practice, the input cost of PAL+SC is not exactly equal to that of PAL as $C$,
> > > as we need to send mini-batch requests for API calls due to token rate limits for each API call.
> > > Specifically, we used a batch size of $5$. So the cost should be $NC/5$ in practice.
> > >
> > > > for OpenAI APIs this is not feasible yet, so the cost needs to be clarified separately
> > >
> > > We appreciate your concern about the cost of APIs.
> > > We will make a new Table 1 to clarify both the input and output API costs separately from the costs from open-source models.
> > >
> > > > why are most results of PAL-SC missing in your main-text Table 1
> > >
> > > Thank you for pointing this out.
> > > For prior works, we only filled in results reported in their works, so the performance of most tasks is missing.

---

> > > > ### Comment · Reviewer_fpKX · 2023-08-15
> > > >
> > > > Thanks for showing the two tables with the actual costs. They make the comparison more straightforward. I'd encourage including these tables in the new version for more transparent discussions of the proposed approach.
> > > >
> > > > > And we choose `x_B=600` as the maximum length following the implementation code of PAL. The max `T` was set as `10` in our method
> > > >
> > > > Estimating the cost with "maximum length" does not make sense to me as one can set an arbitrarily large max. The actual cost or using an average length is more practical.
> > > >
> > > > > It is worth noting that in practice, the input cost of PAL+SC is not exactly equal to that of PAL as , as we need to send mini-batch requests for API calls due to token rate limits for each API call.
> > > >
> > > > Thanks for clarifying it!

---

### Official Review · Reviewer_pGEM · 2023-07-07

**Soundness:** 3 good
**Presentation:** 3 good
**Contribution:** 3 good
**Rating:** 6
**Confidence:** 4

**Summary:**

The authors propose a self-evaluation guided strategy to improve the rationale decoding process from CoT sampling. They also explore ways to balance the quality of evaluation with diversity of generation. Overall, their approach work well on existing datasets and improve few-shot accuracy on multiple math and commonsense datasets. Additionally, their approach leads to more robust reasoning.

**Strengths:**

* The ability to guide the rationale generation process via self quality evaluation and temperature-controlled randomness is an effective solution towards improving the decoding time performance and consistency of CoT sampling process
* The work is timely in the community since CoT is a predominant way to generate rationales using LLMs. This approach can help the effort in improving robustness of these rationales


**Weaknesses:**

* Since the stepwise self-evaluation involves multiple calls to the base LLM, this adds an overall overhead both in terms of compute and cost. There is no discussion regarding this in the paper. This restricts the reader to understand the feasibility of such an approach. Please include some time and cost analysis with and without the self-evaluation based search. I understand that the authors point some of these in the limitations, but I strongly believe they are crucial to have in the paper.
* Can you consider results on more recent models that do provide logits (say GPT3) to showcase the utility of this approach even in the more recent models? This is optional, but a good to have analysis on a few datasets (say StrategyQA and GSM8K).

**Questions:**

* How are the different hyper-parameters ($\alpha, \lambda$, etc.) tuned? Or do you use some generic settings for all datasets?

---

> ### Author Rebuttal · Authors · 2023-08-10
>
> We thank the reviewer for the helpful suggestions!
>
> > Please include some time and cost analysis with and without the self-evaluation based search
>
> Sure. Please see the global response for this.
>
> For API call time costs, we have proposed an implementation to accelerate the inference.
> Our method averages $6$ minutes per instance for single-chain reasoning, compared to the $4$ minutes taken by PAL+SC.
> We provide detailed procedures with the implementation code in the supplementary materials.
>
>
> > Can you consider results on more recent models that do provide logits (say GPT3)
>
> In the original paper, we utilize Codex (`code-davinci-002`) as the LLM backbone to report our main results.
> Codex actually performs significantly better than GPT-3.5 (InstructGPT -- `text-davinci-003`).
> The comparisons between Codex and GPT-3.5 are available in the PAL paper (Figures 7 and 8).
>
> We agree with the reviewer on adding more recent models that provide logits.
> We run additional experiments on LLaMA 2 as a representative of open-source models.
> We illustrate the preliminary results in Table 2 of the attached pdf.
> We will add the complete LLaMA 2 results to the next revision of the paper.
>
>
> > How are the different hyper-parameters tuned? Or do you use some generic settings for all datasets?
>
> Most of the hyperparameters are the same for all datasets.
> For example, we set $\lambda=0.5$, beam size $k=5$, number of rollouts $n=16$, and the temperature decay ratio $\alpha=0.5$.
> We only tune the sampling temperature for each dataset using the corresponding dev set, since easier and more objective tasks require a higher temperature for more diversified samples.
> Please refer to Appendix A.2 and A.3 for a detailed breakdown.

---

> > ### Comment · Reviewer_pGEM · 2023-08-11
> > **Response to authors**
> >
> > Thanks for clarifying my concerns. I appreciate adding results for the latest LLaMA 2 model. Given the latest comment (https://openreview.net/forum?id=Bw82hwg5Q3&noteId=ZmCBVKeGdD), I'm understanding that this approach is severely computationally expensive if done for single CoT ($20k$ vs $k$) and moderately expensive for SC ($160k$ vs $40k$). Given this significant limitation, I agree with Reviewer WznY that the results in Table1 are bit misleading as it does not paint the complete picture. You should modify the table (and potentially main claims) to reflect how the method compares in terms of accuracy with a comparable compute budget across both methods. Without this, the gains are not correctly attributed. Having said that, I think this is not a major limitation of the method. Your approach has a larger impact on performance with increasing computation cost, something that SC cannot potentially match, even if we allow it to generate more chains. But this needs to be depicted+discussed more clearly in the paper, which is currently lacking. Until this is addressed appropriately, I'm reducing my scores.

---

> > > ### Author Response · Authors · 2023-08-14
> > >
> > > > You should modify the table (and potentially main claims) to reflect how the method compares in terms of accuracy with a comparable compute budget across both methods
> > >
> > > We appreciate your concerns about the cost analysis. Please refer to the [comment](https://openreview.net/forum?id=Bw82hwg5Q3&noteId=b85ueeEcFP) under the global response for details.
> > >
> > > > Having said that, I think this is not a major limitation of the method. Your approach has a larger impact on performance with increasing computation cost, something that SC cannot potentially match, even if we allow it to generate more chains.
> > >
> > > Thanks for the insightful feedback. Our latest experiments with open-source models in the [comment](https://openreview.net/forum?id=Bw82hwg5Q3&noteId=b85ueeEcFP) under the global response have indeed shown this phenomenon as you pointed out.

---

> > > > ### Comment · Reviewer_pGEM · 2023-08-18
> > > > **Response to authors**
> > > >
> > > > Thanks for adding these new experiments. They are helpful in understanding the benefits of this approach under equal compute constraints. Please do include them in the final version as agreed and modify the introduction+results section to emphasize these findings. Additionally, it will help to increase this analysis to some more datasets (potentially some non-math dataset as well). I'm increasing my score to reflect my new understanding of the work based on these experiments.

---

### Author Rebuttal · Authors · 2023-08-10

We appreciate your comments and suggestions.
Below are our responses to the concerns about cost.

We acknowledge that beam search incurs a significant cost.

- In light of reviewers' concern about cost, we conducted a detailed analysis about the cost as shown in Table 1 and Table 2 in the attached pdf, for scenarios using both APIs and open-sourced models. If APIs are used in the multiple chain case, a significant amount of computation is spent on processing identical few-shot prompts for each generation, leading our method to cost $O(n \cdot T)$ times more in computation, where $n$ is the number of rollouts for each beam and $T$ is the number of reasoning steps. While repetitive API calls can lead to increased input prompt costs, these costs can be also largely mitigated by caching past hidden states during inference if we use open-sourced models. In this context, our method's cost is $O(n)$ times greater than the PAL + self-consistency baseline.


- In reasoning, methods are typically compared using the same number of reasoning chains for self-consistency (Wang et al. 2023; Zhou et al. 2023). As such, we aimed to achieve the best performance under the same number of reasoning chains.
To find a balance between performance and computational cost, we can decrease $n$, the number of rollouts for each beam. By reducing $n$ from $16$ to $8$, the performance slightly drops from $80.2$ to $79.9$, but it is still a substantial improvement over the baseline performance of $72.0$.


- Our method leads to a correct and interpretable reasoning chain while self-consistency only improves the accuracy of the final answer.

We will include the detailed analysis and experiments in our updated version.

---
Self-Consistency Improves Chain of Thought Reasoning in Language Models (Wang et al. 2023)

Least-to-Most Prompting Enables Complex Reasoning in Large Language Models (Zhou et al. 2023)

---

> ### Author Response · Authors · 2023-08-14
>
> We appreciate the reviewers' comments on performance comparison under comparable computation costs.
> Here we present the complete results noting the cost and compare with the baselines under similar costs.
>
> To strike a balance between effectiveness and computational cost, we set the number of rollouts per beam candidate $n$, to $2$ in the following experiments.
> In our original submission, we did not control the computation cost and adopted a relatively large $n$ ($n=16$). Here we present a more complete picture.
> Specifically, we base our new experiments on LLaMA2-7B and LLaMA2-13B since Codex is not available and ChatGPT API does not provide probabilities.
>
> In the following tables, we show results on GSM8K with equivalent computational costs (\#Tokens per instance) using alphabets A-F.
> We use the actual \#Tokens to represent the cost, as the sum of the generated tokens (\#Gen. Tokens) and the additional input tokens for self-evaluation per instance.
> Here $N$ denotes the number of members in majority voting. $k$ and $n$ represent the beam size and the number of rollouts per beam candidate.
>
> **LLaMA 13B**
>
> |  Methods  |  \#Tokens  |  \#Gen. Tokens  |   N   |   k   |   n   |     Accuracy     |Cost|
> |    :-:    |    :-:    |       :-:      | :-:   | :-:   | :-:   |        :-:       |:-:|
> |  PAL      |   339.9   |     339.9      |   -   |   -   |   -   |        33.3      |
> |   PAL+SC  |   1.0k    |     1.0k       |   3   |   -   |   -   |      35.8        | A |
> |  Ours     |   1.1k    |     748.3      |   -   |   2   |   2   |        36.4      | A |
> |           |           |                |       |       |       |                  |   |
> | PAL+SC    |   3.0k    |     3.0k       |   9   |   -   |   -   |     39.4         | B |
> |  Ours     |   2.8k    |     1.9k       |   -   |   5   |   2   |        41.7      | B |
> |           |           |                |       |       |       |                  |   |
> | PAL+SC    |   5.8k    |     5.8k       |   17  |   -   |   -   |       43.9       | C |
> |  Ours+SC  |   5.6k    |     3.8k       |  10   |   5   |   2   |     45.5         | C |
> |           |           |                |       |       |       |                  |   |
> | PAL+SC    |   8.8k    |     8.8k       |   26  |   -   |   -   |        47.0      | D |
> |  Ours+SC  |   8.4k    |     2.2k       |  15   |   5   |   2   |       50.0       | D |
> |           |           |                |       |       |       |                  |   |
> | PAL+SC    |    17k    |      17k       |  50   |   -   |   -   |       48.4       | E |
> |  Ours+SC  |  16.8k    |    11.4k       |  30   |   5   |   2   |        54.7      | E |
> |           |           |                |       |       |       |                  |   |
> | PAL+SC    |    30k    |      17k       |  90   |   -   |   -   |       48.5       | F |
> |  Ours+SC  |    28k    |      19k       |  50   |   5   |   2   |        55.6      | F |
>
> **LLaMA 7B**
>
> |  Methods  |  #Tokens  |  #Gen. Tokens  |   N   |   k   |   n   |     Accuracy     |Cost|
> |    :-:    |    :-:    |       :-:      | :-:   | :-:   | :-:   |        :-:       |:-:|
> |  PAL      |   346.0   |     346.0      |   -   |   -   |   -   |        19.7      |
> |  Ours     |   1.1k    |     723.2      |   -   |   5   |   2   |        24.2      |
> |  PAL+SC   |   3.4k    |     3.4k       |  10   |   -   |   -   |        24.0      | A |
> |  Ours+SC  |   3.3k    |     2.2k       |  15   |   5   |   2   |        28.0      | A |
> |           |           |                |       |       |       |                  |   |
> |  PAL+SC   |   6.8k    |     6.8k       |  20   |   -   |   -   |           28.8   | B |
> |  Ours+SC  |   6.6k    |     3.6k       |  30   |   5   |   2   |       34.8       | B |
> |           |           |                |       |       |       |                  |   |
> |  PAL+SC   |    17k    |      17k       |  50   |   -   |   -   |     31.8         | C |
> |  Ours+SC  |    11k    |     7.2k       |  50   |   5   |   2   |      39.4        | C |
> |           |           |                |       |       |       |                  |   |
> |  PAL+SC   |    28k    |      28k       |  80   |   -   |   -   |     31.1         | D |
> |  Ours+SC  |    18k    |      12k       |  80   |   5   |   2   |      41.7        | C |
>
> **Key Observations**
>
> - Our method consistently outperforms self-consistency when benchmarked for equal computational cost (measured in \#tokens).
> For example, we observe absolute increase in accuracy of $2.3$\%,  $1.6$\%, $3.0$\%, $6.3$\%, and $7.1$\% respectively on the five cost levels using LLaMA-13b.
>
> - On the other hand, self-consistency saturates at around $N=50$.
>
> Thanks for highlighting the importance of comparison based on computation cost.
> We will include comprehensive experiments and discussions on this in the revised version.
> We will also reorganize the main result tables to provide a clear and holistic image of the results and the computation costs.

---

> > ### Author Response · Authors · 2023-08-14
> >
> > For reference, our ablation analysis on LLaMA 2 (13b) shows that reducing $n$ to as low as $2$ can reduce computational costs while maintaining similar performance.
> >
> > |  Methods  |   N   |   k   |   n   |         Accuracy     |
> > |   :-:     |  :-:  |  :-:  |  :-:  |            :-:       |
> > |  Ours     |   -   |   5   |   2   |     41.7             |
> > |  Ours     |   -   |   5   |   4   |      42.4            |
> > |  Ours     |   -   |   5   |   6   |           40.9       |

---

### Decision · Program_Chairs · 2023-09-21

**Decision:**

Accept (poster)

**Comment:**

This paper proposes an approach to integrate self-evaluation and decoding through stochastic beam search. Authors show that self-evaluation guided stochastic beam search provides a reasonable trade-off between quality and diversity and improves the performance on several well-known reasoning tasks. Reviewers had several concerns regarding the experiment details and fair comparison with prior work which was addressed by authors to a large extent (thanks to authors' great job during the rebuttal period). The main issue is that the amount of changes are significant and the experiment section could benefit from a complete rewrite. That makes this a difficult decision for me. After discussing with reviewers, my final recommendation is to accept the paper. However, I'm urging the authors to take time to rewrite the experiment section, making sure they address reviewers' concerns (and in particular reviewer WznY).